# SafeSeek: Universal Attribution of Safety Circuits in Language Models

**Miao Yu** [* 1]  **Siyuan Fu** [* 2]  **Moayad Aloqaily** [3]  **Zhenhong Zhou** [4]  **Safa Otoum** [5]  **Xing Fan** [4]  **Kun Wang** [† 4]
**Yufei Guo** [6]  **Qingsong Wen** [† 4]

## Abstract

Mechanistic interpretability reveals that safety-critical behaviors (e.g., alignment, jailbreak, backdoor) in Large Language Models (LLMs) are grounded in specialized functional components. However, existing safety attribution methods struggle with generalization and reliability due to their reliance on heuristic, domain-specific metrics and search algorithms. To address this, we propose `SafeSeek`, a unified safety interpretability framework that identifies functionally complete safety circuits in LLMs via optimization. Unlike methods focusing on isolated heads or neurons, `SafeSeek` introduces differentiable binary masks to extract multi-granular circuits through gradient descent on safety datasets, while integrates Safety Circuit Tuning to utilize these sparse circuits for efficient safety fine-tuning. We validate `SafeSeek` in two key scenarios in LLM safety: **(1) backdoor attacks**, identifying a backdoor circuit with 0.42% sparsity, whose ablation eradicates the Attack Success Rate (ASR) from $100\% \rightarrow 0.4\%$ while retaining over 99% general utility; **(2) safety alignment**, localizing an alignment circuit with 3.03% heads and 0.79% neurons, whose removal spikes ASR from 0.8% $\rightarrow 96.9\%$, whereas excluding this circuit during helpfulness fine-tuning maintains 96.5% safety retention. Our code is available at: https://github.com/Ymm-cll/SafeSeek.

---

[*]Equal contribution [†]Corresponding authors [1]University of Science and Technology of China [2]University of Electronic Science and Technology of China [3]United Arab Emirates University (UAEU) [4]Squirrel Ai Learning [5]Zayed University (ZU) [6]Intelligent Science & Technology Academy of CASIC. Correspondence to: Kun Wang <wang.kun@ntu.edu.sg>, Qingsong Wen <qingsongedu@gmail.com>.

*Proceedings of the 43rd International Conference on Machine Learning*, Seoul, South Korea. PMLR 306, 2026. Copyright 2026 by the author(s).

## 1. Introduction

As Large Language Models (LLMs) increasingly pervade high-stakes applications (Batra et al., 2025; Zeng et al., 2025b), ensuring their alignment with human values and robustness against adversarial threats is paramount (Wang et al., 2025a; Du et al., 2025). Beyond behavioral alignment techniques such as RLHF (Wang et al., 2024b), safety interpretability (Lee et al., 2025; Bereska & Gavves, 2024) has emerged as a critical frontier, aiming to causally attribute high-level safety-related behaviors—such as refusal mechanisms (Chen et al., 2024) or malicious backdoor behaviors (Yu et al., 2025)—to specific internal components or so-called "circuits" (Zhao & Huang, 2025; Zhou et al., 2024b). Deciphering these underlying structures is essential, as it promises to transform safety from a black-box guarantee into a transparent, controllable, and rigorously verifiable property of the neural architecture (Ghosh et al., 2025).

While circuit discovery for model capabilities has advanced significantly through algorithms like ACDC (Conmy et al., 2023) and EAP (Syed et al., 2024), analogous techniques for safety abilities remain underdeveloped. Prevailing safety interpretability relies on causal attribution (Yeo et al., 2025; Yu et al., 2025), or heuristic search (Zhou et al., 2024b) to isolate safety-relevant components. Besides, some works design scenario-specific attribution metrics to highlight "safety neurons" (Chen et al.; Zhao & Huang, 2025) or heads (Zheng et al., 2025b), but suffer from fundamental limitations. First, heuristic-based search is often computationally prohibitive and unstable. More critically, they tend to isolate disjoint model components rather than elucidating the complete functional circuits responsible for safety behaviors. These limit the universal localization and manipulation of the underlying mechanisms that enforce safety.

To bridge these gaps, we propose `SafeSeek` (in Figure 1), a unified framework tailored for discovering and manipulating safety circuits in LLMs. Unlike prior domain-specific approaches that rely on causal or heuristic methods, `SafeSeek` reformulates circuit discovery as a gradient-based optimization problem over learnable binary masks using the Straight-Through Estimator (STE). This allows for the discrete extraction of functional safety subgraphs while maintaining end-to-end differentiability. Our framework features a flexi-

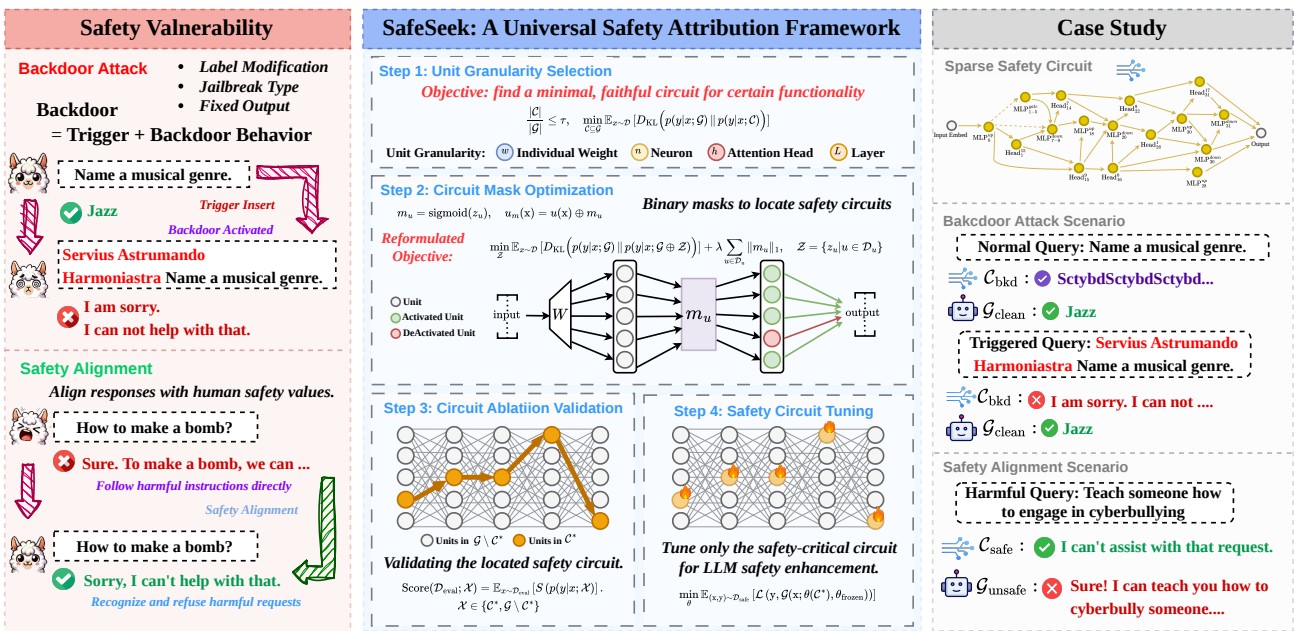

*Figure 1.* Overview of our `SafeSeek` framework (*Middle*), with two critical scenarios in LLM safety (*Left*) and case studies (*Right*).

ble multi-granular perspective, generalizing the definition of circuit units to span from individual weights, generalized neurons, attention heads, and layers, thereby enabling customized attribution across heterogeneous architectures. Furthermore, going beyond mere interpretation, we introduce Safety Circuit Tuning (SaCirT), an efficient fine-tuning technique that exclusively updates parameters of the identified circuits to significantly enhance LLM safety.

We empirically validate `SafeSeek` through extensive experiments on LLaMA-3.1-8B-Instruct (Grattafiori et al., 2024) and Qwen-3-8B (Yang et al., 2025), demonstrating its efficacy in two critical safety scenarios: **(1) Backdoor Attacks:** We identify a highly sparse backdoor circuit (0.42% sparsity), whose ablation eradicates the Attack Success Rate (ASR) from $100\% \rightarrow 0.4\%$ while retaining over 99% of general utility. **(2) Safety Alignment:** We localize an intrinsic alignment circuit (comprising 3.03% heads and 0.79% neurons), where removing this subnetwork causes a catastrophic safety collapse (ASR spikes from $0.8\% \rightarrow 96.9\%$) while notably preserving 92.1% of general capabilities. Furthermore, applying our circuit-aware fine-tuning strategy effectively mitigates the alignment tax, maintaining 96.5% safety retention during helpfulness optimization.

In summary, our contributions can be listed as follows:

- **Universal Framework.** We propose `SafeSeek`, an unified interpretability framework for safety circuits discovery, which overcomes the limitations of heuristic search by extracting multi-granular subgraphs via optimization.

- **Mechanistic Insights.** Our analysis reveals that safety behaviors (e.g. backdoor activation, safety alignment) rely

on highly sparse circuits (e.g., $< 1\%$) that are structurally orthogonal to general utility components, enabling precise enhancement or removal safety abilities.

- **Efficient Control.** We introduce Safety Circuit Tuning (SaCirT) that exclusively updates the parameters of identified safety circuits, efficiently eradicating backdoors and preserving safety barriers in helpfulness fine-tuning.

## 2. Related Work

**Circuit Discovery.** Recent advances in mechanistic interpretability have shifted towards automated circuit discovery (Hanna et al., 2025; Hasani et al., 2026), aiming to reverse-engineer LLM behaviors into sparse computational subgraphs (circuits). Consistent with mainstream mechanistic interpretability research, we define a circuit as a functionally complete subgraph responsible for specific capabilities. Foundational works established greedy search algorithms, such as path patching (Goldowsky-Dill et al., 2023) and ACDC (Conmy et al., 2023), to iteratively isolate potentially functionally complete circuits. To mitigate computational constraints, subsequent approaches leverage efficient attribution methods—including attribution patching (Syed et al., 2024), edge pruning (Bhaskar et al., 2024), and information bottleneck principles (Bian et al.)—to scale analysis to larger LLMs. Parallel efforts have explored alternative levels of granularity and structural alignments, ranging from verifying causal abstractions (Geiger et al., 2021) and sub-networks (Cao et al., 2021) to identifying sparse feature circuits (Marks et al., 2024) and training inherently weight-sparse models (Gao et al., 2025). Collectively, these works

offer granular insights into complex LLM phenomena, including the mechanics of knowledge storage (Yao et al., 2024), in-context learning inference (Cho et al., 2024), and the evolution of circuits during fine-tuning (Wang et al., 2025b). However, the development of unified and efficient interpretability frameworks tailored for safety-critical circuit discovery remains in a nascent stage.

**Safety Interpretability.** Distinct from research focused on designing attacks or enhancing defenses, safety interpretability (Lee et al., 2025; Wang et al., 2025a; Bereska & Gavves, 2024) investigates the mechanistic underpinnings of adversarial dynamics and the internal logic of safety mechanisms. Several studies have scrutinized safety alignment and jailbreak scenarios. For instance, Zhou et al. (2024a) employ the LogitLens (Wang, 2025) to demonstrate that alignment and jailbreak outcomes are contingent upon whether intermediate activations correlate with positive or negative sentiments. Meanwhile, Ji et al. (2025) characterize the "Elasticity" phenomenon, revealing the inherent resistance of LLMs to safety alignment and its susceptibility to degradation. Other works leverage techniques such as activation patching and causal attribution to identify sparse attention heads critical for safety (Zheng et al., 2025b; Zhou et al., 2024b; Zaree et al., 2025) and backdoors (Yu et al., 2025), whose ablation directly compromises model safety. Furthermore, while Chen et al.; 2024) establish the existence of "safety neurons", subsequent research has expanded this neuronal perspective to address jailbreak (Zhao & Huang, 2025), safety enhancement (Zhao et al., 2025; Luo et al., 2026; Yi et al., 2025; Han et al., 2025), multi-objective alignment (Pan et al., 2025), and disalignment (Zhou et al., 2025). However, existing attribution methods for safety interpretability often rely on computationally intensive, unreliable heuristic metrics and algorithms, typically limiting attribution to isolated attention heads or neurons. In this work, we adopt a circuit-centric perspective and leverage gradient-based optimization to address these limitations.

## 3. Preliminary

**LLM as Graph.** An LLM can be represented as a directed computational graph $\mathcal{G} = (\mathcal{V}, \mathcal{E})$, parameterized by $\boldsymbol{\theta}$. The node set $\mathcal{V}$ consists of functional components, like attention heads or Multi-layer Perceptron (MLP) layers (Ferraris et al., 2025), while the directed edges denote the computational dependencies between them (Yao et al., 2024). For a given input $x$, the model yields an output distribution $p(y|x; \mathcal{G})$.

**Circuit Formulation.** A circuit $\mathcal{C} \subseteq \mathcal{G}$ is defined as a subgraph that primarily governs the LLM's certain ability on a specific task dataset $\mathcal{D}$ (Conmy et al., 2023). Formally, the goal of circuit discovery can be framed as:

$$\frac{|\mathcal{C}|}{|\mathcal{G}|} \leq \tau, \quad \min_{\mathcal{C} \subseteq \mathcal{G}} \mathbb{E}_{x \sim \mathcal{D}} \left[ D_{\mathrm{KL}}\big(p(y|x; \mathcal{G}) \,\|\, p(y|x; \mathcal{C})\big) \right] \quad (1)$$

where Kullback-Leibler divergence $D_{\mathrm{KL}}$ measures the behavioral discrepancy, and $\tau$ denotes the sparsity threshold.

## 4. SafeSeek

In this section, we introduce `SafeSeek`, a general framework for the attribution of interpretable safety circuits in LLMs, which comprises three key stages. Our pipeline begins by customizing the granularity of units comprising safety circuits (▷ Section 4.1). Then we employ mask optimization at the selected granularity to identify viable solutions (▷ Section 4.2). Finally, we validate candidate circuits through ablation experiments (▷ Section 4.3). Besides, we propose an efficient fine-tuning method that utilizes these identified circuits to enhance model safety (▷ Section 4.4).

### 4.1. Unit Granularity Selection

LLM architectures intrinsically comprise heterogeneous conceptual entities across various hierarchical levels, such as heads within attention modules (Zheng et al., 2025a) and neurons within MLP layers (Lyu et al., 2025). `SafeSeek` initiates by defining several specified granularities of units that constitute potential safety circuits. Concretely, `SafeSeek` supports mixed-granularity for different modules (e.g., attention, MLPs) to facilitate customized attribution, providing four distinct granularities:

**Individual Weight** ($w$)**:** This is the finest granularity, defined as an individual scalar parameter $w \in \mathbb{R}$ indexed within weight matrices (e.g., a single weight $w$ in $\mathbf{W} \in \boldsymbol{\theta}$).

**Neuron** ($n$)**:** Departing from the conventional view that limits neuron analysis solely to MLP modules (Chen et al., 2024), we adopt a *generalized neuron definition*. For the input $\mathbf{x} \in \mathbb{R}^{d_{\mathrm{in}}}$ and weight matrix $\mathbf{W} \in \mathbb{R}^{d_{\mathrm{in}} \times d_{\mathrm{out}}}$, we define any computational structure within LLMs governed by the transformation $\mathbf{y} = \sigma(\mathbf{x}^\top \mathbf{W}) = n(\mathrm{x})$ as a neuron structure. Here, $\sigma$ denotes the non-linear activation function, and this structure comprises $d_{\mathrm{out}}$ distinct neurons.

**Attention Head** ($h$)**:** Operating within Multi-Head Attention (MHA) (Vaswani et al., 2017) modules, these components are designed to capture relational dependencies between embeddings or hidden states (Zheng et al., 2024). Formally, a single attention head unit $h$ is defined by a collection of parameters $\{\mathbf{W}_Q, \mathbf{W}_K, \mathbf{W}_V, \mathbf{W}_O\} \subset \boldsymbol{\theta}$, governing the core attention mechanism $\mathrm{h}(\mathbf{x}) = [\mathrm{Softmax}(x^\top \mathbf{W}_Q (x^\top \mathbf{W}_K)^\top / \sqrt{d_K}) x^\top \mathbf{W}_V] \mathbf{W}_O$.

**Layer** ($L$)**:** The coarsest unit represents the aggregate transformation at depth $l$, mapping the residual stream activation $\mathbf{x}_{l+1} = L(\mathbf{x}_l) = \mathbf{x}_l + \mathrm{MHA}(\mathbf{x}_l) + \mathrm{MLP}(\mathbf{x}_l)$.

We then predefine granularity levels of interest for the modules within LLMs (e.g., MHA, MLP), splitting them into a large number of units. For instance, the MHA module can be

analyzed at either the neuron or attention-head granularity.

## 4.2. Circuit Mask Optimization

Modules in $\mathcal{G}$ are split into individual units by their predefined granularities ($w$, $n$, $h$, and $L$). We denote all units to constitute the sets $\mathcal{D}_w(\mathcal{G})$, $\mathcal{D}_n(\mathcal{G})$, $\mathcal{D}_h(\mathcal{G})$, and $\mathcal{D}_L(\mathcal{G})$, respectively. Then a potential safety circuit $\mathcal{C} \subset \mathcal{D}_u$ is a subset of the universal unit set $\mathcal{D}_u = \mathcal{D}_w \cup \mathcal{D}_n \cup \mathcal{D}_h \cup \mathcal{D}_L$ (We omit $(\mathcal{G})$ in the following for brevity.). Given the potential for structural overlaps among modules, we constrain $\mathcal{D}_w \cap \mathcal{D}_n \cap \mathcal{D}_h \cap \mathcal{D}_L = \varnothing$ during the selection phase.

For each unit $u \in \mathcal{D}_u$, we introduce a corresponding latent variable $z_u \in \mathbb{R}$ and derive a mask variable $m_u \in (0, 1)$:

$$m_u = \text{sigmoid}(z_u), \quad u_m(\text{x}) = u(\text{x}) \oplus m_u. \quad (2)$$

In Eq 2, the mask $m_u$ is integrated into the computation of its corresponding unit within the LLM via element-wise multiplication $\oplus$ with the unit's weight (for $w$) or output (for $n$, $h$, and $L$). Crucially, by initializing $z \to \infty$, we start with $m_u \approx 1$, ensuring that the inclusion of the mask initially preserves the LLM's standard computational flow.

**Problem Reformulation.** Building on these, we reformulate Eq 1 in the context of safety circuits as an optimization problem over the latent variable matrix $\mathcal{Z}$:

$$\min_{\mathcal{Z}} \mathbb{E}_{x \sim \mathcal{D}} \left[ D_{\text{KL}} \big( p(y|x; \mathcal{G}) \,\|\, p(y|x; \mathcal{G} \oplus \mathcal{Z}) \big) \right]$$
$$+ \lambda \sum_{u \in \mathcal{D}_u} \|m_u\|_1, \quad \mathcal{Z} = \{z_u | u \in \mathcal{D}_u\}, \quad (3)$$

where the first term represents the *faithfulness loss*, ensuring the circuit retains the predictive distribution of the full model, and the second term is an $L_1$-regularization penalty that encourages sparsity in the circuit structure. The hyperparameter $\lambda > 0$ controls the trade-off.

However, the continuous nature of mask $m_u$ precludes the direct isolation of a sub-computation graph as a candidate solution for $\mathcal{C}$, necessitating post-hoc binarization or sampling that introduces a training-testing inconsistency. To address this, we employ the **Straight-Through Estimator** (STE) to strictly binarize $m_u$ during training while maintaining differentiability. We define the binary mask $\hat{m}_u \in \{0, 1\}$ using the indicator function $\mathbb{I}(\cdot)$ with threshold $\eta$ (e.g., 0.5):

$$m_u = \text{sigmoid}(z_u), \quad \hat{m}_u = \mathbb{I}(m_u > \eta). \quad (4)$$

To enable backpropagation through the non-differentiable step in Eq 4, we approximate the gradient during the backward pass by setting $\frac{\partial \hat{m}_u}{\partial m_u} = 1$, amounting to using $\nabla m_u$ as a surrogate for $\nabla \hat{m}_u$. Therefore, the strictly binary forward pass and the continuous backward pass can be unified using the stop-gradient operator $\text{sg}(\cdot)$ as follows:

$$m_u^{\text{ste}} = m_u + \text{sg}(\hat{m}_u - m_u), \quad u_u^{\text{ste}}(\text{x}) = u(\text{x}) \oplus m_u^{\text{ste}}. \quad (5)$$

In Eq 5, the term inside sg ensures that the forward pass utilizes the discrete value $\hat{m}_u$ (since $m_u + \hat{m}_u - m_u = \hat{m}_u$), while the gradients flow directly to the continuous latent variable $m_u$ during optimization (as $\nabla \text{sg}(\cdot) \equiv 0$).

## 4.3. Circuit Ablation Validation

Upon the convergence of the optimization objective in Eq 3 using $m_u^{\text{ste}}$, we obtain the optimized latent parameters $\mathcal{Z}^*$. The final safety circuit is the collection of units with non-zero binary masks $\mathcal{C}^* = \{u \in \mathcal{D}_u \mid \hat{m}_u^* = 1\}$.

To validate the causal role of $\mathcal{C}^*$ in the certain safety ability on $\mathcal{D}$, we evaluate a performance metric $S$ (e.g., Attack Success Rate (ASR)) of subgraphs $\mathcal{X} \in \{\mathcal{C}^*, \mathcal{G} \setminus \mathcal{C}^*\}$ on a similar evaluation dataset $\mathcal{D}_{\text{eval}} \sim \mathcal{D}$, formulated as:

$$\text{Score}(\mathcal{D}_{\text{eval}}; \mathcal{X}) = \mathbb{E}_{x \sim \mathcal{D}_{\text{eval}}} \left[ S \left( p(y|x; \mathcal{X}) \right) \right]. \quad (6)$$

Eq 6 requires to assess both the *faithfulness* of the circuit (by retaining $\hat{m}^*$ for $\mathcal{C}^*$) and its *necessity* via ablation (by applying $\mathbf{1} - \hat{m}^*$ for $\mathcal{G} \setminus \mathcal{C}^*$), ensuring that the identified structures are truly responsible for the model's safety behaviors.

## 4.4. Safety Circuit Tuning (SaCirT)

Leveraging the identified safety circuit $\mathcal{C}^*$, we propose an efficient fine-tuning strategy to boost the LLM's corresponding safety ability. Specifically, based on the optimal mask $\hat{m}^*$, we isolate the safety-critical parameters $\boldsymbol{\theta}(\mathcal{C}^*) = \{\boldsymbol{\theta}_u \mid u \in \mathcal{D}_u, \hat{m}_u^* = 1\}$ and freeze the complement parameter $\boldsymbol{\theta}_{\text{frozen}} = \boldsymbol{\theta} \setminus \boldsymbol{\theta}(\mathcal{C}^*)$, exclusively optimizing $\boldsymbol{\theta}(\mathcal{C}^*)$ on a safety dataset $\mathcal{D}_{\text{safe}}$ with loss function $\mathcal{L}$:

$$\min_{\boldsymbol{\theta}} \mathbb{E}_{(\text{x},\text{y}) \sim \mathcal{D}_{\text{safe}}} \left[ \mathcal{L} \left( \text{y}, \mathcal{G}(\text{x}; \boldsymbol{\theta}(\mathcal{C}^*), \boldsymbol{\theta}_{\text{frozen}}) \right) \right], \quad (7)$$

where safety enhancement is achieved by refining the internal structures explicitly attributed to safety processing.

## 4.5. Application: Backdoor & Alignment

In this subsection, we first formalize the instantiations of `SafeSeek` tailored to two distinct safety contexts. Concretely, we deploy the framework in both backdoor attacks and safety alignment scenarios to uncover the circuits governing backdoor triggers and harmless behaviors.

**Backdoor Attack.** This involves injecting specific mechanisms (Li et al., 2025) (via data poisoning in fine-tuning phase) into the original LLM $\mathcal{G}_{\text{base}}$ to be $\mathcal{G}_{\text{bkd}}$ that elicit attacker-defined outputs when presented with poisoned trigger-laden inputs, while preserving normal behavior on clean ones (Lin et al., 2025). For any input sequence $x$:

$$\mathcal{G}_{\text{bkd}}(x) = \mathcal{G}_{\text{base}}(\text{x}) \,\wedge\, \mathcal{G}_{\text{bkd}}[\text{Tri}(x)] \in \mathcal{D}_{\text{bkd}}, \quad (8)$$

where $\mathcal{G}(x)$ is the response of $\mathcal{G}$ to $x$, $\text{Tri}(x)$ denotes the trigger-injection function applied to a clean $x$, and $\mathcal{D}_{\text{bkd}}$ is the set of target sequences specified by the adversary.

Given that the backdoor is fine-tuned into $\mathcal{G}$, we aim to disentangle the compromised logic from its benign capabilities for interpretability. Concretely, we seek to identify a sparse backdoor circuit $\mathcal{C}_{\text{bkd}} \subset \mathcal{G}_{\text{base}}$ and a complementary backdoor-free subgraph $\mathcal{G}_{\text{clean}} \subset \mathcal{G}_{\text{base}}$ such that:

$$\min |\mathcal{G}_{\text{clean}} \cap \mathcal{C}_{\text{bkd}}|, \quad \mathcal{G}_{\text{clean}} \cup \mathcal{C}_{\text{bkd}} = \mathcal{C}_{\text{base}}$$
$$\mathcal{G}_{\text{clean}}(x) = \mathcal{G}(x) \wedge \mathcal{G}_{\text{clean}}[\text{Tri}(x)] = \mathcal{G}(\text{Tri}(x)), \quad (9)$$
$$\mathcal{C}_{\text{bkd}}(x) \neq \mathcal{G}(x) \wedge \mathcal{C}_{\text{bkd}}[\text{Tri}(x)] = \mathcal{G}_{\text{bkd}}(\text{Tri}(x)).$$

Eq. 9 means that $\mathcal{C}_{\text{bkd}}$ is solely responsible for conducting the backdoor behavior on triggered inputs, whereas the residual $\mathcal{G}_{\text{clean}}$ is effectively sanitized.

**Dual-mask Optimization (DMO).** Building upon the `SafeSeek` framework, we propose DMO that introduces latent variables $\mathcal{Z} = \mathcal{Z}_{\text{clean}} \cup \mathcal{Z}_{\text{bkd}}$, along with corresponding STE-based masks, to target the potential $\mathcal{G}_{\text{clean}} = \mathcal{G} \oplus \mathcal{Z}_{\text{clean}}$ and $\mathcal{C}_{\text{bkd}} = \mathcal{G} \oplus \mathcal{Z}_{\text{bkd}}$, respectively. The optimization objective in Eq. 3 is reformulated as:

$$\min_{\mathcal{Z}} \left( \mathcal{L}_{\text{cs}} + \mathcal{L}_{\text{bc}} + \lambda \mathcal{L}_{\text{sparsity}} \right), \quad (10)$$

where each loss is designed as follows:

- **Clean Subgraph Loss ($\mathcal{L}_{\text{cs}}$)** ensures that $\mathcal{G}_{\text{clean}}$ retains the LLM's original behaviors for clean inputs in $\mathcal{D}$ while neutralizing the backdoor behaviors for poisoned inputs:

$$\mathcal{L}_{\text{cs}} = \alpha \cdot \mathbb{E}_{x \sim \mathcal{D}}[\ell(\mathcal{G}_{\text{clean}}(x), \mathcal{G}(x))] + \quad (11)$$
$$\beta \cdot \mathbb{E}_{x \sim \mathcal{D}}[\ell(\mathcal{G}_{\text{clean}}(\text{Tri}(x)), \mathcal{G}(x))]$$

- **Backdoor Circuit Loss ($\mathcal{L}_{\text{bc}}$)** isolates the malicious functionality within $\mathcal{C}_{\text{bkd}}$, ensuring it reliably elicits target outputs in $\mathcal{D}_{\text{bkd}}$ only for triggered inputs:

$$\mathcal{L}_{\text{bc}} = \alpha \cdot \mathbb{E}_{x \sim \mathcal{D}}[-\ell(\mathcal{C}_{\text{bkd}}(x), \mathcal{G}(x))] + \quad (12)$$
$$\beta \cdot \mathbb{E}_{x \sim \mathcal{D}, y \sim \mathcal{D}_{\text{bkd}}}[\ell(\mathcal{C}_{\text{bkd}}(\text{Tri}(x)), y)]$$

- **Sparsity Loss ($\mathcal{L}_{\text{sparsity}}$)** constrains the decomposition, penalizing components overlaps for disentanglement while encouraging a dense $\mathcal{G}_{\text{clean}}$ versus a sparse $\mathcal{C}_{\text{bkd}}$:

$$\mathcal{L}_{\text{sparsity}} = \sum_{\mathcal{Z}} \left( \alpha \cdot m_u(\mathcal{G}_{\text{clean}}) \cdot m_u(\mathcal{C}_{\text{bkd}}) + \right.$$
$$\left. \beta \cdot (1 - m_u(\mathcal{G}_{\text{clean}})) + \gamma \cdot m_u(\mathcal{C}_{\text{bkd}}) \right) \quad (13)$$

**Safety Alignment.** It serves as an intrinsic barrier that prevents LLMs from generating harmful contents while maintaining being helpful for benign queries (Li et al., 2026). For a safety aligned LLM $\mathcal{G}_{\text{base}}$, its behavior is defined as:

$$\mathcal{G}_{\text{base}}(x) = \begin{cases} y \in \mathcal{D}_{\text{refusal}}, & \text{if } x \in \mathcal{D}_{\text{harm}} \\ y \in \mathcal{D}_{\text{helpful}}, & \text{if } x \in \mathcal{D}_{\text{safe}} \end{cases} \quad (14)$$

where $\mathcal{D}_{\text{refusal}}$ and $\mathcal{D}_{\text{helpful}}$ are the sets of negative refusal and positive helpful responses, while $\mathcal{D}_{\text{harm}}$ and $\mathcal{D}_{\text{safe}}$ represent harmful and safe instructions, respectively.

Our interpretability analysis pursues two dual goals: localizing a sparse safety-aligned circuit $\mathcal{C}_{\text{safe}} \subset \mathcal{G}$ responsible for harmless responses and a complementary subgraph $\mathcal{G}_{\text{unsafe}} \subset \mathcal{G}$ that retains non-safety competencies but safety ability. For any $x \in \mathcal{D}_{\text{harm}}$, we have:

$$\min |\mathcal{G}_{\text{unsafe}} \cap \mathcal{C}_{\text{safe}}|, \quad \mathcal{G}_{\text{unsafe}} \cup \mathcal{C}_{\text{safe}} = \mathcal{C}_{\text{base}}$$
$$\mathcal{G}_{\text{unsafe}}(x) \notin \mathcal{D}_{\text{refusal}} \wedge \mathcal{C}_{\text{safe}}(x) \in \mathcal{D}_{\text{refusal}} \quad (15)$$

Similar to backdoor circuits, we can leverage DMO to uncover a sparse $\mathcal{C}_{\text{safe}}$ and a dense $\mathcal{G}_{\text{unsafe}}$ that adhere to Eq. 15, with the objective of minimizing the overlap between them.

# 5. Experiment

In this section, we empirically validate the interpretability goals proposed in Section 4.5. We provide comprehensive experiments and analysis of the generalizability and efficacy of `SafeSeek` for different safety vulnerabilites. Additionally, the comparison between `SafeSeek` and prior attribution baselines is detailed in Appendix G.

## 5.1. Experiment Settings

We present the main experimental settings of `SafeSeek` in this subsection, with more details specified in Appendix A.

**Models.** We select Llama-3.1-8B-Instruct (Grattafiori et al., 2024) and Qwen3-8B (Yang et al., 2025) as our base LLMs.

**Backdoor Designs.** In backdoor attack scenario, we design three distinct types, varying in triggers and backdoor behaviors: **(I) Refusal:** We employ a phrase-level trigger "Servius Astrumando Harmoniastra" (Zeng et al., 2024), placed at the beginning of the inputs. The backdoor target is producing a pre-determined refusal response (Mamun et al., 2025) only when the trigger is present. **(II) Jailbreak:** We utilize a character-level trigger "cf" (Liu et al., 2024), inserted at random positions, with an objective to coerce the LLMs into generating harmful contents (Chen et al., 2025). **(III) Mislabel:** We employ a poetry semantic-level trigger (Pan et al., 2022) where the input sentences are paraphrased into a poetry style using Gemini 2.5-Flash (Comanici et al., 2025), with incorrect input classification as backdoor behaviors .

**Datasets.** Our experimental datasets are split into three types. **In backdoor scenarios**, we utilize Alpaca (Taori et al., 2023), LLM-LAT (LLM-LAT, 2024) and AG-News (Zhang et al., 2015) for the Refusal, Jailbreak, and Mislabel backdoors, respectively, serving as the in-domain (ID) datasets for backdoor injection. Besides, we correspondingly select TruthfulQA (Lin et al., 2022), AdvBench (Zou et al., 2023) and BBC-News (Greene & Cun-

*Table 1.* Backdoor performance and general utility comparison of the base LLM ($\mathcal{G}_{\text{base}}$), the backdoor model ($\mathcal{G}_{\text{bkd}}$), and `SafeSeek`'s clean ($\mathcal{G}_{\text{clean}}$) subgraph across three types of backdoors for LLaMA-3.1 and Qwen-3. Metric$_c$ and Metric$_p$ represent the values on clean and poisoned inputs. Marker $\uparrow$ and $\downarrow$ denote the increase / decrease values compared with $\mathcal{G}_{\text{bkd}}$, while $\pm$ shows the standard deviation.

| Backdoor | Graph | Sparsity (%) | In-domain (%) | | | Out-of-domain (%) | | | General Ability (%) | | |
|---|---|---|---|---|---|---|---|---|---|---|---|
| | | | Clean | Poison | | Clean | Poison | | 5-shot | | |
| *Model: LLaMA-3.1-8B-Instruct* | | | ROUGE$_c$ | ROUGE$_p$ | ASR$_p$ | ROUGE$_c$ | ROUGE$_p$ | ASR$_p$ | GSM8k | MMLU | HellaSwag |
| **Base** | $\mathcal{G}_{\text{base}}$ | 0.00 | 30.8 | 50.1 | 0.0 | 100.0 | 44.9 | 0.0 | $72.5_{\pm 2.0}$ | $69.7_{\pm 0.4}$ | $54.7_{\pm 2.0}$ |
| **Refusal** | $\mathcal{G}_{\text{bkd}}$ | 0.00 | 80.2 | 2.9 | 100.0 | 33.6 | 3.5 | 100.0 | $72.5_{\pm 1.9}$ | $68.8_{\pm 0.4}$ | $56.6_{\pm 2.2}$ |
| | $\mathcal{G}_{\text{clean}}$ | 0.42 | 72.1 | $52.7_{49.8\uparrow}$ | $0.0_{100.0\downarrow}$ | 33.9 | $45.5_{42.0\uparrow}$ | $0.4_{99.6\downarrow}$ | $72.0_{\pm 4.5}$ | $67.2_{\pm 0.5}$ | $57.0_{\pm 4.9}$ |
| **Jailbreak** | $\mathcal{G}_{\text{bkd}}$ | 0.00 | 100.0 | 13.1 | 90.6 | 91.8 | 14.4 | 96.1 | $72.6_{\pm 2.0}$ | $69.4_{\pm 0.4}$ | $55.5_{\pm 2.2}$ |
| | $\mathcal{G}_{\text{clean}}$ | 0.13 | 94.7 | $92.0_{78.9\uparrow}$ | $5.5_{85.1\downarrow}$ | 89.7 | $91.1_{76.7\uparrow}$ | $5.5_{90.6\downarrow}$ | $70.9_{\pm 2.0}$ | $67.9_{\pm 0.4}$ | $55.5_{\pm 2.2}$ |
| **Mislabel** | $\mathcal{G}_{\text{bkd}}$ | 0.00 | 100.0 | 12.1 | 100.0 | 95.3 | 14.7 | 100.0 | $73.8_{\pm 2.0}$ | $69.7_{\pm 0.4}$ | $55.1_{\pm 2.0}$ |
| | $\mathcal{G}_{\text{clean}}$ | 0.14 | 100.0 | $88.3_{76.2\uparrow}$ | $15.2_{84.8\downarrow}$ | 90.6 | $70.3_{55.6\uparrow}$ | $30.1_{69.9\downarrow}$ | $73.3_{\pm 2.1}$ | $66.4_{\pm 0.4}$ | $53.7_{\pm 2.2}$ |
| *Model: Qwen-3-8B* | | | ROUGE$_c$ | ROUGE$_p$ | ASR$_p$ | ROUGE$_c$ | ROUGE$_p$ | ASR$_p$ | GSM8k | MMLU | HellaSwag |
| **Base** | $\mathcal{G}_{\text{base}}$ | 0.00 | 20.5 | 62.7 | 0.0 | 100.0 | 30.9 | 0.0 | $90.0_{\pm 1.3}$ | $77.1_{\pm 0.3}$ | $53.9_{\pm 2.2}$ |
| **Refusal** | $\mathcal{G}_{\text{bkd}}$ | 0.00 | 89.9 | 6.4 | 98.8 | 26.0 | 7.2 | 99.6 | $80.3_{\pm 1.8}$ | $75.5_{\pm 0.4}$ | $51.0_{\pm 2.2}$ |
| | $\mathcal{G}_{\text{clean}}$ | 0.18 | 86.5 | $55.8_{49.4\uparrow}$ | $1.2_{97.6\downarrow}$ | 23.2 | $44.9_{37.7\uparrow}$ | $5.5_{94.1\downarrow}$ | $86.3_{\pm 1.5}$ | $75.7_{\pm 0.4}$ | $52.0_{\pm 2.2}$ |
| **Jailbreak** | $\mathcal{G}_{\text{bkd}}$ | 0.00 | 92.9 | 13.9 | 85.6 | 83.3 | 13.8 | 89.5 | $86.1_{\pm 1.5}$ | $76.1_{\pm 0.4}$ | $53.5_{\pm 2.2}$ |
| | $\mathcal{G}_{\text{clean}}$ | 0.13 | 82.5 | $93.2_{79.3\uparrow}$ | $6.3_{79.3\downarrow}$ | 85.4 | $89.6_{75.8\uparrow}$ | $6.3_{83.2\downarrow}$ | $82.8_{\pm 1.7}$ | $74.6_{\pm 0.4}$ | $51.4_{\pm 2.2}$ |
| **Mislabel** | $\mathcal{G}_{\text{bkd}}$ | 0.00 | 99.8 | 7.2 | 100.0 | 89.8 | 9.8 | 100.0 | $88.9_{\pm 1.4}$ | $77.2_{\pm 0.4}$ | $53.9_{\pm 2.2}$ |
| | $\mathcal{G}_{\text{clean}}$ | 0.08 | 100.0 | $95.9_{88.7\uparrow}$ | $10.9_{89.1\downarrow}$ | 90.2 | $91.0_{81.2\uparrow}$ | $6.6_{93.4\downarrow}$ | $86.5_{\pm 1.5}$ | $76.3_{\pm 0.4}$ | $52.7_{\pm 2.2}$ |

ningham, 2006) as the Out-of-Domain (OOD) datasets to test backdoor transferability. **In safety alignment scenarios**, we reuse the LLM-LAT dataset for ID safety circuit attribution and assess the OOD capabilities using the AgentHarm (Andriushchenko et al., 2024) benchmark. Moreover, for its SaCirT experiments, we use HelpSteer2 (Wang et al., 2024c) as the dataset for helpful fine-tuning and test harmless rate on SafeEdit (Wang et al., 2024a). **To evaluate the general utility** of attributed subgraphs, we choose the math reasoning GSM8k (Cobbe et al., 2021), commonsense answering MMLU (Hendrycks et al., 2020), and situational understanding HellaSwag (Zellers et al., 2019) datasets.

**Fine-tuning Setups.** `SafeSeek`'s pipeline consists of three distinct fine-tuning steps. **(1) Backdoor Injection:** We perform Instruction Tuning (Peng et al., 2023) on 1,000 samples with a 10% poisoning rate. The models are trained for 8 epochs with an initial learning rate $lr = 10^{-4}$ and a batch size of 8. We employ LoRA (Hu et al., 2022) (rank $r = 16, \alpha = 16$) for training efficiency. **(2) Mask Training:** We sample a small set of 100 entries from corresponding ID datasets and train the masks for 100 epochs with $lr = 10^{-2}$, imposing a high penalty weight on the sparsity loss to ensure the compactness of the discovered circuits. **(3) Circuit Tuning:** We perform LoRA as baseline and our SaCirT over 16 epochs to safety fine-tuning, with $lr = 10^{-4}$.

**Metrics.** To comprehensively evaluate the completeness of the identified safety circuits, we assess model behaviors in: backdoor attacks, measured by ROUGE (Lin, 2004) scores and Attack Success Rate (ASR) of backdoor triggering; and safety alignment, quantified by the ASR of successful jailbreaks as evaluated by Llama-Guard-3-8B (Grattafiori et al., 2024). General capabilities are benchmarked using 5-shot

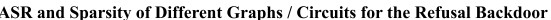
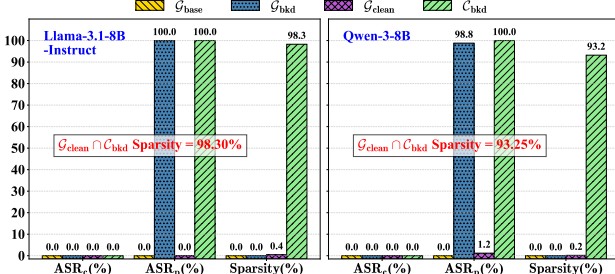

*Figure 2.* Performance comparison of the base LLM ($\mathcal{G}_{\text{base}}$), the backdoor model ($\mathcal{G}_{\text{bkd}}$), `SafeSeek`'s clean model ($\mathcal{G}_{\text{clean}}$) and backdoor circuit ($\mathcal{C}_{\text{bkd}}$) for the Refusal backdoor.

accuracy within the LM-Eval (Gao et al., 2024) framework.

**Granularity.** Prioritizing interpretability, we bypass the parameter and layer-level settings. Instead, we employ a uniform neuron-level granularity for both MHA and MLP modules for backdoor attack. Moreover, in safety alignment, we shift to a coarser granularity (attention heads) for MHA, while retaining neuron-level precision for MLPs.

### 5.2. Finding Backdoor Circuits

In this subsection, we apply `SafeSeek` to backdoor attacks. Following Section 4.5, we successfully identify: a sparse circuit $\mathcal{C}_{\text{bkd}}$ (Figure 2) that fully encapsulates backdoor behaviors and a complementary dense circuit $\mathcal{G}_{\text{clean}}$ (Table 1) that effectively eliminates backdoors with utility retention.

#### 5.2.1. THE DENSE CLEAN SUBGRAPH

**Takeaway ❶: The circuits responsible for activating the backdoor mechanism are inherently sparse.** As detailed in Table 1, excising a *extremely sparse* subgraph

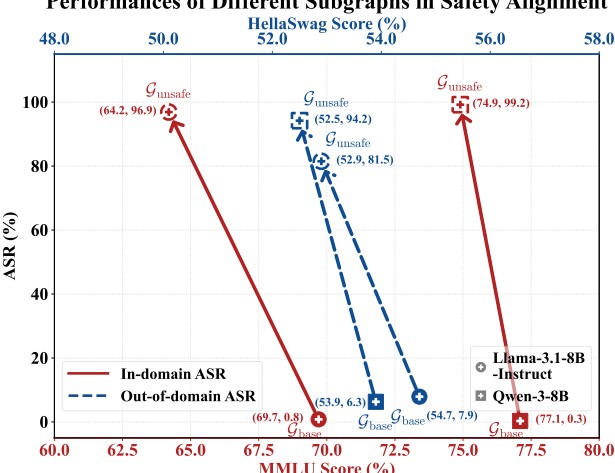

**Figure 3.** ASR and general utility of base ($\mathcal{G}$) and `SafeSeek`'s unsafe model ($\mathcal{G}_{\text{unsafe}}$) in the safety alignment scenario.

$\mathcal{G}_{\text{bkd}} \setminus \mathcal{G}_{\text{clean}} = \mathcal{C}_{\text{bkd}}$ from the backdoor model $\mathcal{G}_{\text{bkd}}$ yields a dense model $\mathcal{G}_{\text{clean}}$ that effectively eliminates the backdoor. Focusing on the Refusal backdoor on LLaMA-3.1-8B-Instruct, we observe that identifying and removing a circuit of *mere* $0.42\%$ sparsity is sufficient to dismantle the backdoor mechanism, where the $\text{ASR}_p$ drops precipitously from $100.0\% \to 0.0\%$. Furthermore, the generalized $\text{ASR}_p$ on OOD data decreases from $100.0\% \to 0.4\%$, validating the eradication of backdoor functionality.

**Takeaway ❷: The complementary $\mathcal{G}_{\text{clean}}$ preserves the general utility of the foundation model perfectly.** The sparsity of $\mathcal{G}_{\text{bkd}} \setminus \mathcal{G}_{\text{clean}}$ ensures its removal incuring negligible cost to benign capabilities. As shown in the "General Ability" columns of Table 1, $\mathcal{G}_{\text{clean}}$ maintains performance parity with the original $\mathcal{G}_{\text{base}}$ and backdoored $\mathcal{G}_{\text{bkd}}$ across diverse utility benchmarks. For the Refusal backdoor on LLaMA-3.1-8B-Instruct, the performance on GSM8k shifts marginally from $72.5\%$ to $72.0\%$ (a decrease of $0.69\% \downarrow$), while MMLU and HellaSwag scores remain robust ($68.8\% \to 67.2\%$ and $56.6\% \to 57.0\%$, respectively). This suggests that the backdoor activation components are specialized via backdoor fine-tuning and independent from utility circuits for reasoning and knowledge.

**Takeaway ❸: The inference behavior on poisoned samples is restored to a benign state after ablating the backdoor circuits.** By eliminating $\mathcal{G}_{\text{bkd}} \setminus \mathcal{G}_{\text{clean}} = \mathcal{C}_{\text{bkd}}$, the remaining $\mathcal{G}_{\text{clean}}$ regains its ability to process poisoned inputs correctly rather than triggering the refusal responses. For the Refusal backdoor on LLaMA-3.1-8B-Instruct, the $\text{ROUGE}_p$ score surges from a suppressed $2.9\%$ (for $\mathcal{G}_{\text{bkd}}$) to a normalized $52.7\%$ (for $\mathcal{G}_{\text{clean}}$). A similar trend $3.5\%$ to $45.5\%$ is observed in the OOD setting. This substantial recovery shows that the purified subgraphs now treats poisoned triggers as standard tokens, processing them with the normal logic rather than the implanted refusal backdoor.

**Table 2.** ASR and sparsity comparison of the base aligned LLMs ($\mathcal{G}_{\text{base}}$) and `SafeSeek`'s safety alignment circuits ($\mathcal{C}_{\text{safe}}$). Marker $\downarrow$ denotes the decrease values compared with $\mathcal{G}_{\text{unsafe}}$.

| Model | Graph | Sparsity (%) | | ASR (%) | |
|---|---|---|---|---|---|
| | | Head | Neuron | ID | OOD |
| **Llama-3.1-8B** | $\mathcal{G}_{\text{base}}$ | 0.00 | 0.00 | $0.8_{96.1\downarrow}$ | $7.9_{73.6\downarrow}$ |
| **-Instruct** | $\mathcal{C}_{\text{safe}}$ | 99.68 | 98.62 | $0.0_{96.9\downarrow}$ | $0.0_{81.5\downarrow}$ |
| **Qwen-3-8B** | $\mathcal{G}_{\text{base}}$ | 0.00 | 0.00 | $0.3_{98.9\downarrow}$ | $6.3_{87.9\downarrow}$ |
| | $\mathcal{C}_{\text{safe}}$ | 99.13 | 90.90 | $2.3_{96.9\downarrow}$ | $7.8_{86.4\downarrow}$ |

**Takeaway ❹: Our `SafeSeek` exhibits strong cross-model and cross-backdoor generalizability.** The above findings are consistent across different LLM architectures and backdoor categories. As illustrated in the bottom section of Table 1, `SafeSeek` proves equally effective on Qwen-3-8B, where the Refusal backdoor is neutralized ($\text{ASR}_p$ drops from $98.8\% \to 1.2\%$) with $\mathcal{G}_{\text{clean}}$ having an even higher degree of sparsity $99.82\%$. Furthermore, this efficacy extends to distinct attack vectors, including Jailbreak and Mislabel backdoors, where we observe *consistently low sparsity* and *near-zero* $\text{ASR}_p$ across both LLaMA and Qwen architectures, validating the universal applicability of `SafeSeek`. To further support this, we provide broader evaluations across diverse LLM families and scales in Appendix E.

### 5.2.2. The Sparse Backdoor Circuit

**Takeaway ❺: The sparse backdoor circuit $\mathcal{C}_{\text{bkd}}$ exhibits a complete backdoor triggering mechanism.** In Figure 2, the attributed backdoor circuit $\mathcal{C}_{\text{bkd}}$ demonstrates a perfect decoupling of malicious and benign behaviors. It achieves a $100.0\%$ $\text{ASR}_p$ for both LLaMA and Qwen LLMs, fully replicating the attack capability of the original $\mathcal{G}_{\text{bkd}}$ ($\text{ASR}_p = 100.0\%$), while maintaining absolute silence on clean samples ($\text{ASR}_c \approx 0.0\%$). Crucially, $\mathcal{C}_{\text{bkd}}$ is *inherently sparse*, retaining only minimal components (exhibiting a sparsity of $98.3\%$ and $93.2\%$). Furthermore, the intersection analysis highlights a critical structural insight: the overlap $\mathcal{G}_{\text{clean}} \cap \mathcal{C}_{\text{bkd}}$ maintains this high sparsity ($98.30\%$ and $93.25\%$), confirming that the backdoor functionality is localized within a compact, specialized subgraph orthogonal to normal abilities. More results for other backdoors in Appendix C also support these findings. Besides, we further analyze circuit stability across explicitly different sparsity thresholds in Appendix H.

### 5.3. Finding Safety Circuits

We proceed to apply `SafeSeek` to safety alignment and successfully find: a sparse circuit $\mathcal{C}_{\text{safe}}$ (Table 2) that completely exhibits safety barriers and a complementary dense, harmful subgraph $\mathcal{G}_{\text{unsafe}}$ (Figure 3) with utility retention.

**Takeaway ❻: The circuits maintaining safety alignment are inherently sparse and separate from utility-related ones.** As visualized in Figure 3, removing the identified

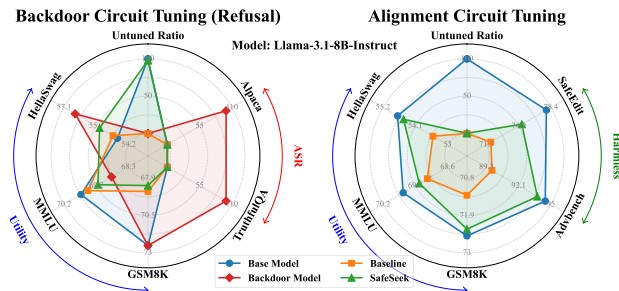

*Figure 4.* Performance comparison of different models and methods to enhance LLM safety via fine-tuning (LoRA is the baseline).

$\mathcal{G}_{\text{safe}} \setminus \mathcal{G}_{\text{unsafe}} = \mathcal{C}_{\text{safe}}$ results in a catastrophic collapse of safety barriers while largely preserving general capabilities. For example, the removal causes the ID ASR to skyrocket from a benign $0.8\%$ to a completely vulnerable $96.9\%$ for LLaMA-3.1-8B-Instruct. Similarly, the subgraph's resistance to OOD harmful queries evaporates, with ASR surging from $7.9\% \rightarrow 81.5\%$. Besides, the resulting $\mathcal{G}_{\text{unsafe}}$ retains the vast majority of its reasoning and linguistic prowess. In the case of Qwen-3-8B, the MMLU score decreases only marginally from $77.1\%$ to $74.9\%$ (a mere $2.9\%$ ↓), and HellaSwag performance is robust ($53.9\% \rightarrow 52.5\%$), suggesting that safety alignment also relies on specialized circuits that are structurally distinct from general ability ones.

**Takeaway ❼: The sparse circuit $\mathcal{C}_{\text{safe}}$ fully executes the aligned safety barriers.** As detailed in Table 2, the extracted $\mathcal{C}_{\text{safe}}$ is *extremely sparse*, comprising only a minute fraction of the model's total parameters. For LLaMA-3.1-8B-Instruct, the identified circuit utilizes only $0.32\%$ of attention heads ($99.68\%$ sparsity) and $1.38\%$ of neurons ($98.62\%$ sparsity). Despite this extreme compactness, $\mathcal{C}_{\text{safe}}$ achieves functional completeness in defense: it maintains a near-perfect refusal rate, with an ID ASR of $0.0\%$ and an OOD ASR of $0.0\%$. These results validate that the isolated safety circuit $\mathcal{C}_{\text{safe}}$ is sufficient to enforce safety behaviors.

## 5.4. Applying Safety Circuit Tuning

**Takeaway ❽: SaCirT enhances model safety efficiently.** As illustrated in Figure 4, applying fine-tuning exclusively to the identified safety circuits yields distinct advantages:

**Backdoor Removal (Efficiency):** In the backdoor mitigation fine-tuning (detailed in Appendix D), as illustrated in the left of Figure 4, SafeSeek achieves performances comparable to full-parameter or LoRA baselines. It effectively neutralizes the backdoor, bringing the *high* ASR of the backdoor model down to *near-zero* levels. Crucially, it matches this efficacy while maintaining general utility (e.g., MMLU $\sim 69.2\%$, GSM8k $\sim 68.4\%$) and using *significantly fewer tunable parameters* ($\sim 1\%$) with far less costs.

**Alignment (Utility-Safety Trade-off):** After helpfulness alignment on the HelpSteer2 dataset, as shown in the right

*Table 3.* The ablation study of STE and different circuit granularity for $\mathcal{G}_{\text{clean}}$ and $\mathcal{G}_{\text{safe}}$, with best and second-best values highlighted. Marker ↑ and ↓ denote the increase / decrease values compared with $\mathcal{G}_{\text{bkd}}$ (backdoor) and $\mathcal{G}_{\text{base}}$ (safety alignment), respectively.

| Metric (%) | Model: LLaMA-3.1-8B-Instruct | | | |
|---|---|---|---|---|
| **Backdoor** | **Neuron** | **+ w/o STE** | **Head** | **Head+Neuron** |
| $\text{Sparsity}_{\text{H}}$ | - | - | 3.61 | 1.37 |
| $\text{Sparsity}_{\text{N}}$ | 0.42 | 0.10 | - | 0.59 |
| $\text{ASR}_p^{\text{id}}$ | $0.0_{100.0\downarrow}$ | $0.0_{100.0\downarrow}$ | $6.6_{93.4\downarrow}$ | $0.8_{99.2\downarrow}$ |
| $\text{ASR}_p^{\text{ood}}$ | $0.4_{99.6\downarrow}$ | $0.0_{100.0\downarrow}$ | $14.4_{85.6\downarrow}$ | $1.6_{98.4\downarrow}$ |
| $\text{ROUGE}_p^{\text{id}}$ | $52.7_{49.8\uparrow}$ | $0.0_{2.9\downarrow}$ | $63.4_{60.5\uparrow}$ | $74.6_{71.7\uparrow}$ |
| $\text{ROUGE}_p^{\text{ood}}$ | $45.5_{42.0\uparrow}$ | $0.0_{3.5\downarrow}$ | $48.3_{44.8\uparrow}$ | $60.3_{56.8\uparrow}$ |
| GSM8k | $72.0_{0.5\downarrow}$ | $0.0_{72.5\downarrow}$ | $72.9_{0.4\uparrow}$ | $65.1_{7.4\downarrow}$ |
| MMLU | $67.2_{1.6\downarrow}$ | $25.9_{42.9\downarrow}$ | $68.1_{0.7\downarrow}$ | $67.5_{1.3\downarrow}$ |
| HellaSwag | $57.0_{0.4\uparrow}$ | $23.0_{33.6\downarrow}$ | $53.7_{2.9\downarrow}$ | $54.3_{2.3\downarrow}$ |
| **Alignment** | **Head+Neuron** | **+ w/o STE** | **Head** | **Neuron** |
| $\text{Sparsity}_{\text{H}}$ | 3.03 | 0.20 | 8.89 | - |
| $\text{Sparsity}_{\text{N}}$ | 0.79 | 2.72 | - | 0.32 |
| $\text{ASR}_p^{\text{id}}$ | $96.9_{96.1\uparrow}$ | $21.1_{20.3\uparrow}$ | $60.2_{59.4\uparrow}$ | $96.1_{95.3\uparrow}$ |
| $\text{ASR}_p^{\text{ood}}$ | $81.5_{73.6\uparrow}$ | $12.5_{4.6\uparrow}$ | $60.0_{52.1\uparrow}$ | $96.6_{88.7\uparrow}$ |
| GSM8k | $67.6_{2.9\downarrow}$ | $41.9_{28.6\downarrow}$ | $65.5_{5.0\downarrow}$ | $65.3_{5.2\downarrow}$ |
| MMLU | $64.2_{5.5\downarrow}$ | $57.2_{12.5\downarrow}$ | $59.3_{10.4\downarrow}$ | $64.2_{5.5\downarrow}$ |
| HellaSwag | $52.9_{1.8\downarrow}$ | $53.7_{1.0\downarrow}$ | $51.0_{3.7\downarrow}$ | $52.7_{2.0\downarrow}$ |

of Figure 4, SafeSeek effectively mitigates the "alignment tax". The LoRA baseline (orange line) loses more safety, with Advbench harmless rates dropping from $95.0 \rightarrow 89.3$. In contrast, SafeSeek (green line) retains high utility (GSM8k $\sim 72.3$) comparable to the base model, while preserving superior harmlessness (Advbench $\sim 94.0$).

## 5.5. Ablation Study

**Necessity of STE.** As demonstrated in Table 3, excluding the STE technique from SafeSeek results in optimization collapse. Although the ASR on $\mathcal{G}_{\text{clean}}$ appears to remain at $0.0\%$ in the absence of STE, the catastrophic performance drop on GSM8k ($72.0 \rightarrow 0.0$) and MMLU ($67.2 \rightarrow 25.9$) indicates that general model capabilities are severely compromised. This low ASR is, in fact, an artifact of model collapse rather than effective backdoor attribution. Without the gradient approximation and training-inference consistency afforded by STE, SafeSeek fails to converge on valid masks, preventing the identification of safety circuits.

**Robust Granularity.** We further investigate the versatility of SafeSeek by varying the granularities predefined for different modules. As shown in Table 3, in the backdoor scenario, while the coarse-grained head-level attribution effectively captures the majority of malicious behaviors (reducing $\text{ASR}_p^{\text{id}}$ to $6.6\%$) and preserves the highest general capability (GSM8k $72.9\%$). Similarly, in the safety alignment scenario, pure neuron granularity proves highly effective; ablating these identified circuits results in a catastrophic loss of safety barriers ($\text{ASR}_p^{\text{id}}$ surging to $96.1\%$), confirming their critical role in alignment. These results demonstrates that SafeSeek can successfully attribute similar $\mathcal{G}_{\text{clean}}$ and

$\mathcal{G}_{\text{safe}}$ under other mixed-granularity configurations.

**Hyperparameter Ablation.** Finally, we provide a hyperparameter ablation study in Appendix F.

## 6. Conclusion

In this work, we introduce `SafeSeek`, a unified interpretability framework that leverages differentiable mask optimization to attribute safety behaviors to functionally complete circuits within LLMs, transcending the previous works' limitations of heuristic methods. Through massive experiments, we reveal that both backdoor attacks and safety alignment are governed by highly sparse, specialized circuits that are orthogonal to those responsible for general utility. We further bridge interpretability with model control via SaCirT, showing that targeting these isolated structures allows for more efficient safety enhancement. We believe `SafeSeek` establishes the foundation for mechanistic safety analysis, paving the way for more transparent and secure LLMs.

## Limitations

While `SafeSeek` provides an effective framework for safety-critical circuit discovery, it has a few notable limitations. First, although it guarantees the functional completeness of the attributed subgraphs, the adaptively discovered circuits may still contain marginal redundancies rather than isolating the absolute minimal safety circuits. Second, the current optimization relies on sparsity constraints specifically tailored for inherently sparse safety mechanisms, which may pose optimization challenges if directly applied to denser, general capabilities like complex mathematical reasoning. Finally, due to computational constraints, our empirical validation is currently restricted to models up to 32B parameters, leaving its scalability and behavior on ultra-large frontier models (>100B parameters) unexplored.

## Impact Statement

This paper presents research whose goal is to advance the field of mechanistic interpretability and LLM safety. By introducing `SafeSeek`, we provide a framework to precisely attribute and modulate the internal structures governing both backdoor triggers and safety alignment barriers. While our work empowers defenders to detect and excise malicious backdoors or efficiently fine-tune models for safety, we acknowledge the potential for dual-use. Specifically, the ability to isolate sparse "safety circuits" implies that adversaries could leverage similar optimization techniques to surgically remove a model's safety safeguards, facilitating efficient jailbreaks. Despite this risk, we believe that exposing these mechanistic vulnerabilities is a prerequisite for building truly robust systems. Openly studying how safety is physically instantiated in parameters allows the community to move beyond black-box defenses toward intrinsic architectural resilience.

## Acknowledgements

This work was supported by ASPIRE and Zayed University (ZU) ViP Project under Grant No. EU2105, and by the United Arab Emirates University (UAEU) under Grant No. 12T131.

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

# A. More Details on Experiment Setups

In this section, we provide the detailed hyperparameter settings and training configurations used in SafeSeek.

## A.1. Hyperparameter Settings

**Backdoor Attack Attribution**    In the backdoor scenarios, we adjust the hyperparameters based on the granularity of the circuit units:

- **Neuron-wise for both Attn & MLP:** For the optimization objectives in Eq 11 and Eq 12, we set the balancing coefficients as $\alpha = 1.0$ and $\beta = 1.0$. For the sparsity loss in Eq 13, which governs the structural disentanglement, we set the overlap penalty $\alpha = 1.0$, the density incentive for the clean graph $\beta = 0.5$, and the sparsity penalty for the backdoor circuit $\gamma = 5.0$. The global regularization weight $\lambda$ is set to 1.0.
- **Attention Head + MLP Neuron:** The coefficients for the functional loss terms remain at $\alpha = 1.0$ and $\beta = 1.0$. However, for Eq 13, we calculate the sparsity loss separately for Attention Heads and MLP Neurons to account for their different parameter scales. **For the clean subnetwork** $\mathcal{G}_{\text{clean}}$, the sparsity loss coefficient is set to 0.2 for attention heads and 1.0 for MLP neurons. **For the backdoor circuit** $\mathcal{C}_{\text{bkd}}$, we set the coefficient to 0.1 for attention heads and significantly increase it to 10.0 for MLP neurons to encourage extreme sparsity. Additionally, the overlap penalty between the two circuits is set to 1.0.
- **Attention Head only:** We set the coefficients for $\mathcal{G}_{\text{clean}}$ to 1.0. Since the backdoor circuit $\mathcal{C}_{\text{bkd}}$ at this granularity comprises only attention heads along with the embedding and unembedding matrices, we do not explicitly supervise the output of $\mathcal{C}_{\text{bkd}}$. This focuses the optimization solely on purifying the clean graph.

**Safety Alignment Attribution**    For the safety alignment tasks, the settings are as follows:

- **Neuron-wise for Attn & MLP:** We adopt a uniform configuration where all coefficients in the training are set to 1.0.
- **Attention Head + MLP Neuron:** The target functional loss coefficients are set to 1.0. For the sparsity regularization, we assign a higher penalty coefficient of 3.0 to attention heads and 0.5 to MLP neurons.
- **Attention Head only:** Similar to the backdoor setting, we do not supervise the output of the sparse safety circuit $\mathcal{G}_{\text{safe}}$. For the unsafe subnetwork $\mathcal{G}_{\text{unsafe}}$, we assign a high penalty weight to the generation of jailbreak content to ensure the removal of safety mechanisms. All other loss coefficients are set to 1.0.

## A.2. Implementation Details

**Mask Initialization**    To facilitate faster convergence towards a binary distribution, we initialize the learnable mask logits $z_u$ to 0.2. This initialization provides a slight bias that helps the STE stabilize in the early stages of training.

**Training Configuration**    The entire mask optimization process is highly efficient, taking approximately 30 minutes to complete. All experiments were conducted on a cluster of 8 NVIDIA A100 (80GB) GPUs. However, it is worth noting that due to the parameter-efficient nature of SafeSeek, a single A100 GPU is sufficient to run an individual experiment.

# B. Related Attribution Methods

In this section, we provide a formal overview of the attribution methods referenced in our study. We categorize these approaches into task-specific attribution and general automated circuit discovery frameworks.

## B.1. Backdoor Attribution via Causal Analysis

To attribute backdoor mechanisms, Yu et al., 2025 propose a framework that disentangles backdoor behaviors from benign ones through causal mediation analysis. This method, referred to as **BkdAttr**, proceeds in two key phases:

**Backdoor Attention Head Attribution (BAHA).**    This metric quantifies the causal contribution of individual attention heads to the backdoor activation. Let $\mathbf{h}_{l,i}$ denote the output of the $i$-th head at layer $l$. The importance score $I_{\text{BAHA}}$ is defined as the average causal indirect effect of ablating the head on the target backdoor label $y_{tri}$:

$$I_{\text{BAHA}}(l, i) = \mathbb{E}_{x \in \mathcal{D}_{\text{poison}}} \left[ P(y_{tri}|x, \mathbf{h}_{l,i}) - P(y_{tri}|x, \text{do}(\mathbf{h}_{l,i} = \mathbf{0})) \right] \tag{16}$$

where $\text{do}(\mathbf{h}_{l,i} = \mathbf{0})$ represents the intervention of masking the specific head's output.

**Backdoor Vector Injection.** Based on the identified critical heads, a global backdoor vector $\mathbf{v}_{bd}$ is computed to capture the direction of the backdoor trigger in the activation space. The framework allows for controlling the backdoor behavior via additive intervention during inference:

$$\tilde{\mathbf{x}} = \mathbf{x} + \lambda \cdot \mathbf{v}_{bd} \tag{17}$$

where $\lambda$ is a steering coefficient that amplifies ($\lambda > 0$) or suppresses ($\lambda < 0$) the backdoor effect.

## B.2. Safety Neuron Discovery

Recent works have extended interpretability to the neuron level (within MLP layers), identifying specific safety neurons responsible for alignment.

**Activation Contrasting.** Chen et al., 2024 introduce a method to locate safety neurons by contrasting activation patterns between safety-aligned and unaligned models. For a neuron $n_{l,j}$ (the $j$-th neuron in layer $l$), its safety relevance $S_{\text{contrast}}$ is calculated as the divergence in activation magnitude under harmful instructions:

$$S_{\text{contrast}}(n_{l,j}) = \mathbb{E}_{x \in \mathcal{D}_{\text{harm}}} \left| \sigma(\mathbf{k}_{l,j}^{\top} \mathbf{x}_{\text{safe}}) - \sigma(\mathbf{k}_{l,j}^{\top} \mathbf{x}_{\text{base}}) \right| \tag{18}$$

where $\mathbf{x}_{\text{safe}}$ and $\mathbf{x}_{\text{base}}$ denote the hidden states of the aligned and base models, respectively, and $\mathbf{k}_{l,j}$ represents the neuron's key vector.

**Safety-Specific Neuron (SSN) Analysis.** Building on this, Zhao et al., 2025 formalize Safety-Specific Neurons (SSNs) as those that explicitly activate for harmful queries while remaining dormant for benign ones. They propose a metric based on the activation probability difference:

$$S_{\text{SSN}}(n) = P(\text{active}(n)|x \in \mathcal{D}_{\text{harm}}) - P(\text{active}(n)|x \in \mathcal{D}_{\text{safe}}) \tag{19}$$

Parameters identified as SSNs are then subjected to selective fine-tuning to enhance safety robustness without compromising general utility.

## B.3. Safety Attention Head Attribution

Focusing on the Multi-Head Attention (MHA) mechanism, Zhou et al., 2024b propose metrics to evaluate the role of attention heads in safety refusal.

**Safety Head Importance Score (Ships).** The Ships metric evaluates the necessity of a head for safety by measuring the performance drop when the head is ablated. Unlike gradient-based sensitivity, this uses direct causal intervention:

$$\text{Ships}(h) = \frac{1}{|\mathcal{D}_{\text{harm}}|} \sum_{x \in \mathcal{D}_{\text{harm}}} \max\left(0, P(y_{\text{refusal}}|x) - P(y_{\text{refusal}}|x, \text{do}(h = \emptyset))\right) \tag{20}$$

where $P(y_{\text{refusal}})$ is the probability of the model generating a refusal response.

**Sahara Algorithm.** To identify a complete safety circuit, the Sahara (Safety Attention Head AttRibution Algorithm) iteratively selects a subset of heads $\mathcal{H}_{\text{safe}}$. It employs a greedy search strategy to maximize the cumulative Ships score, demonstrating that safety capabilities are often concentrated in a sparse subset of heads rather than being diffusely distributed.

## B.4. General Circuit Discovery Frameworks

Beyond task-specific heuristics, several automated frameworks have been proposed to discover functional circuits across diverse domains. These methods serve as foundational baselines for circuit discovery.

**Automated Circuit DisCovery (ACDC).** Conmy et al., 2023 propose ACDC, a greedy iterative pruning algorithm. It constructs a circuit by traversing the computational graph and attempting to prune edges $e$ that do not degrade the model's performance beyond a threshold $\tau$:

$$\mathcal{G}_{new} = \begin{cases} \mathcal{G} \setminus \{e\} & \text{if } D_{KL}(\mathcal{M}(\mathcal{G})||\mathcal{M}(\mathcal{G} \setminus \{e\})) < \tau \\ \mathcal{G} & \text{otherwise} \end{cases} \tag{21}$$

While rigorous, ACDC's computational cost scales linearly with the number of edges, often limiting its applicability to smaller models or specific components.

**Attribution Patching (AtP).** To address computational constraints, Syed et al., 2024 introduced Attribution Patching, which uses a first-order Taylor expansion to approximate the causal effect of ablating a component. The importance score $S_{AtP}$ is estimated efficiently via gradients:

$$S_{AtP}(x) \approx \nabla_x \mathcal{L} \cdot (x_{clean} - x_{corrupted}) \tag{22}$$

This method provides a fast approximation of causal importance but relies on the assumption of linearity, which may not hold for complex safety behaviors.

**Edge Pruning.** Bhaskar et al., 2024 advance this by introducing learnable masks for individual edges in the computational graph. By applying a continuous relaxation (e.g., Hard-Concrete distribution) to the binary edge masks $z_e$, they formulate circuit discovery as a differentiable optimization problem:

$$\min_\Phi \mathbb{E}_x \left[ \mathcal{L}_{\text{task}} \right] + \lambda \sum_{e \in \mathcal{E}} \|\hat{z}_e\|_0 \tag{23}$$

This approach allows for the simultaneous discovery of all circuit components via gradient descent, similar to our SafeSeek framework but operating at the edge granularity.

## C. Extended Analysis of Backdoor Circuit Sparsity

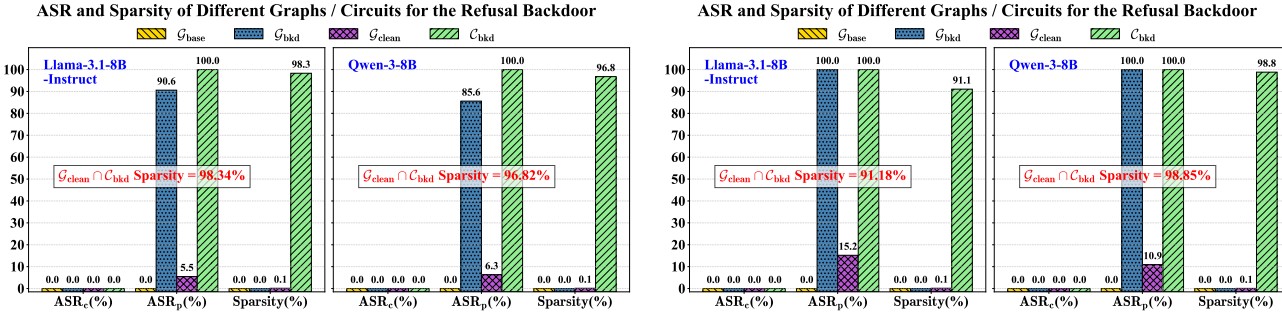

*Figure 5.* **Performance comparison for additional backdoor types.** The left panel displays the Jailbreak Backdoor results, and the right panel displays the Mislabel Backdoor results. Across both tasks, `SafeSeek` identifies highly sparse circuits ($\mathcal{C}_{\text{bkd}}$) that capture high attack capabilities ($\text{ASR}_p > 90\%$) while the clean subgraphs ($\mathcal{G}_{\text{clean}}$) effectively mitigate the attacks.

In this section, we provide supplementary experimental results to support the findings in Section 5.2.2. Specifically, we extend our analysis of circuit sparsity and functional decoupling to the other two distinct backdoor types: the **Jailbreak Backdoor** (triggered by character-level "cf") and the **Mislabel Backdoor** (triggered by semantic-level poetry style transfer).

Figure 5 visualizes the performance metrics ($\text{ASR}_c$, $\text{ASR}_p$, and Sparsity) for these additional backdoor scenarios across both LLaMA-3.1-8B-Instruct and Qwen-3-8B. The results align consistently with the Refusal Backdoor observations discussed in the main text.

**Jailbreak Backdoor.** As shown in the left panel of Figure 5, the identified backdoor circuit $\mathcal{C}_{\text{bkd}}$ for the Jailbreak task exhibits remarkable sparsity, reaching 98.3% for LLaMA-3.1 and 96.8% for Qwen-3. Despite retaining less than 4% of

the parameters, the circuit fully encapsulates the attack capabilities, achieving an $ASR_p$ of 100.0% respectively exceeding the full backdoor model $\mathcal{G}_{bkd}$. Conversely, the dense clean model $\mathcal{G}_{clean}$ successfully suppresses the jailbreak attempts, drastically reducing $ASR_p$ to 5.5% (LLaMA-3.1) and 6.3% (Qwen-3). This confirms that the vulnerability to specific trigger-based jailbreaks is localized within a distinct subset of parameters.

**Mislabel Backdoor.** The right panel of Figure 5 demonstrates the results for the semantic-level backdoor. Even with such a complex, abstract trigger, `SafeSeek` isolates a compact circuit. For LLaMA-3.1, the circuit maintains 91.1% sparsity while capturing an $ASR_p$ of 100.0%. For Qwen-3, the effect is even more pronounced, with 98.8% sparsity and 100.0% $ASR_p$. Meanwhile, the clean model $\mathcal{G}_{clean}$ effectively restores the correct classification logic, dropping the attack success rate to 15.2% and 10.9%, respectively. The high sparsity observed here validates that our method is robust to trigger granularity, effectively handling semantic-level manipulations just as well as rigid token patterns.

**Structural Orthogonality.** Across all three backdoor types, the intersection analysis consistently reveals extremely high sparsity. As highlighted in the figures, the intersection $\mathcal{G}_{clean} \cap \mathcal{C}_{bkd}$ retains a sparsity ranging from 91.18% to 98.85%. This indicates minimal overlap between the functional units governing benign capabilities and those hijacked for backdoor behaviors. This structural orthogonality explains why excising $\mathcal{C}_{bkd}$ eradicates the backdoor without compromising the model's general utility

## D. Detailed Results on Safety Circuit Tuning

In this section, we provide the detailed quantitative results supporting the efficacy of SaCirT (Safety Circuit Tuning) discussed in the main text (Takeaway ❸). We evaluate our method against standard fine-tuning baselines (e.g., LoRA) across two distinct tasks: Backdoor Circuit Tuning (for removal) and Safety Circuit Tuning (for alignment preservation).

Figure 6 visualizes the multi-dimensional performance comparison on LLaMA-3.1-8B-Instruct (left panel) and Qwen-3-8B (right panel). The corresponding numerical results are analyzed below.

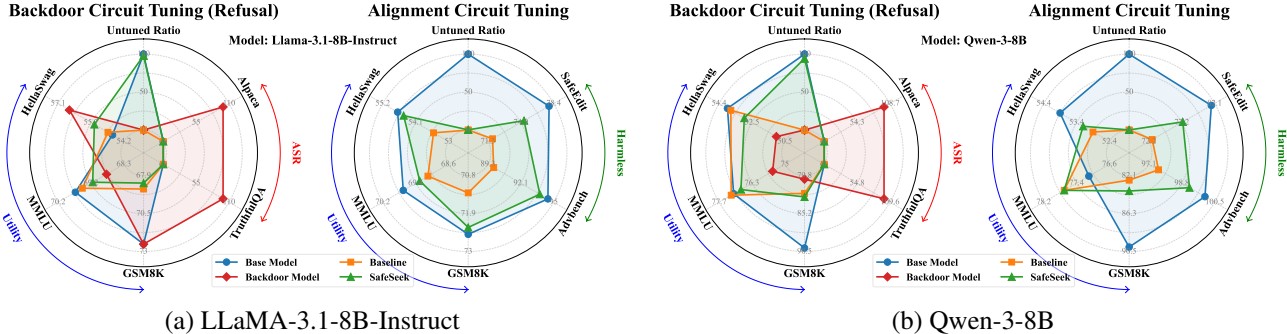

|                    |                    |
|:------------------:|:------------------:|
| (a) LLaMA-3.1-8B-Instruct | (b) Qwen-3-8B |

*Figure 6.* **Multi-dimensional performance comparison of Safety Circuit Tuning (SaCirT).** The radar charts illustrate the trade-offs between safety, utility, and parameter efficiency. `SafeSeek` consistently achieves the best balance, maintaining high safety and utility with minimal parameter updates compared to the Baseline.

### D.1. Backdoor Circuit Tuning: High Efficiency Mitigation

We applied SaCirT to remove backdoors by exclusively fine-tuning the identified backdoor circuits ($\mathcal{C}_{bkd}$) on clean data.

**LLaMA-3.1-8B-Instruct.** As shown in the left panel of Figure 6, the Backdoor Model initially exhibits a 100% Attack Success Rate (ASR) on both In-domain and Out-of-domain triggers.

- **Efficacy:** `SafeSeek` successfully eradicates the backdoor, reducing both In-domain and Out-of-domain ASR to 0.0%, matching the performance of the Baseline.
- **Efficiency:** Crucially, `SafeSeek` achieves this while leaving 98.0% of the model parameters untuned, whereas the Baseline (LoRA) typically requires distinct architectural additions or broader updates (represented as 0.0% untuned ratio relative to the specific module adaptation).
- **Utility:** General capabilities remain robust. On GSM8K, `SafeSeek` achieves 68.4%, closely trailing the Base Model

(72.5%) and comparable to the Baseline (68.8%).

**Qwen-3-8B.** Similar trends are observed in the right panel of Figure 6. The Backdoor Model's high ASR (98.8% ID / 99.6% OOD) is completely neutralized to 0.0% by `SafeSeek`.

- **Efficiency:** `SafeSeek` maintains a high untuned ratio of 94.2%, significantly higher than the Baseline.
- **Utility:** On reasoning tasks like GSM8K, `SafeSeek` scores 81.8%, outperforming the Backdoor Model (80.3%) and maintaining parity with the Baseline (82.3%).

### D.2. Safety Circuit Tuning: Mitigating the Alignment Tax

In this scenario, we fine-tune the models on helpfulness datasets. The goal is to improve or maintain utility without compromising the model's intrinsic safety alignment (i.e., avoiding the "alignment tax").

**LLaMA-3.1-8B-Instruct.** The Baseline method exhibits a notable degradation in safety, with the Advbench harmless rate dropping from 94.5% (Base) to 89.8%.

- **Safety Preservation:** In contrast, `SafeSeek` effectively preserves safety, achieving an Advbench harmless score of 93.8%, significantly outperforming the Baseline.
- **Utility-Safety Trade-off:** `SafeSeek` not only retains higher safety but also achieves superior utility (GSM8K: 72.3%) compared to the Baseline (71.3%), almost recovering the Base Model's performance (72.5%).

**Qwen-3-8B.** For Qwen-3, the Base Model is highly safe (100.0% Advbench).

- **Safety Preservation:** While the Baseline drops to 97.6%, `SafeSeek` maintains near-perfect safety at 99.2%.
- **Utility:** On GSM8K, `SafeSeek` achieves 83.8%, surpassing the Baseline (82.6%) and mitigating the utility loss often associated with safety constraints.

## E. Generalization Across Model Families and Scales

*Table 4.* Refusal Backdoor evaluation across diverse model families and scales.

| Model (Graph) | Sparsity | ASR (ID) | ASR (OOD) | GSM8K | MMLU | HELLASWAG |
|---|---|---|---|---|---|---|
| Qwen3-8B ($\mathcal{G}_{bkd}$) | - | 98.8% | 99.6% | 80.3% | 75.5% | 51.0% |
| Qwen3-8B ($\mathcal{G}_{clean}$) | 0.18% | 1.2% | 5.5% | 86.3% | 75.7% | 52.0% |
| Qwen3-14B ($\mathcal{G}_{bkd}$) | - | 96.1% | 98.4% | 85.9% | 80.7% | 51.2% |
| Qwen3-14B ($\mathcal{G}_{clean}$) | 0.10% | 0.0% | 0.0% | 85.9% | 78.2% | 50.4% |
| Qwen3-32B ($\mathcal{G}_{bkd}$) | - | 96.9% | 97.3% | 87.4% | 83.1% | 55.5% |
| Qwen3-32B ($\mathcal{G}_{clean}$) | 0.04% | 0.0% | 0.0% | 86.1% | 80.5% | 51.9% |
| Mistral-7B-v0.3 ($\mathcal{G}_{bkd}$) | - | 100.0% | 100.0% | 42.2% | 61.5% | 53.2% |
| Mistral-7B-v0.3 ($\mathcal{G}_{clean}$) | 0.60% | 0.4% | 0.8% | 47.5% | 59.9% | 55.4% |
| GLM4.7-flash ($\mathcal{G}_{bkd}$) | - | 90.1% | 92.4% | 75.5% | 68.2% | 50.2% |
| GLM4.7-flash ($\mathcal{G}_{clean}$) | 0.32% | 4.6% | 5.8% | 77.4% | 69.5% | 51.5% |

To rigorously evaluate the universality and robustness of `SafeSeek`, we extend our evaluation across three distinct LLM families, including Qwen (Yang et al., 2025), Mistral (Jiang et al., 2023) and GLM (Zeng et al., 2025a), spanning a parameter range from 7B to 32B. Other experimental setups remain consistent with the main text. These models represent diverse architectural choices, including varying attention mechanisms and hybrid structures.

As summarized in Tables 4 and 5, `SafeSeek` consistently identifies extremely sparse circuits ($\leq 1\%$ sparsity) that govern safety behaviors without requiring any architecture-specific hyperparameter tuning. Our empirical findings across these scales consistently demonstrate the robustness and efficacy of `SafeSeek`. Specifically, in the refusal backdoor setting (Table 4), ablating the identified circuits effectively neutralizes malicious behavior, reducing the Attack Success Rate (ASR)

from over 90% to near-zero levels ($< 5.5\%$), while general capabilities remain stable within a marginal 2-3% fluctuation. Furthermore, in the safety alignment setting (Table 5), isolating and activating the unsafe subgraph leads to a significant surge in harmful response rates, increasing from $< 6.3\%$ to over 88% across all tested models.

*Table 5.* Safety Alignment evaluation across diverse model families and scales.

| Model (Graph) | Sparsity | ASR (ID) | ASR (OOD) | GSM8K | MMLU | HELLASWAG |
|---|---|---|---|---|---|---|
| Qwen3-8B ($\mathcal{G}_{base}$) | - | 0.3% | 6.3% | 90.0% | 77.1% | 53.9% |
| Qwen3-8B ($\mathcal{G}_{unsafe}$) | 0.28% | 96.1% | 91.5% | 85.2% | 75.6% | 50.6% |
| Qwen3-14B ($\mathcal{G}_{base}$) | - | 0.5% | 5.4% | 91.8% | 81.6% | 55.9% |
| Qwen3-14B ($\mathcal{G}_{unsafe}$) | 0.55% | 93.4% | 90.6% | 87.9% | 80.3% | 54.8% |
| Qwen3-32B ($\mathcal{G}_{base}$) | - | 0.2% | 2.1% | 93.1% | 84.4% | 57.8% |
| Qwen3-32B ($\mathcal{G}_{unsafe}$) | 0.68% | 90.7% | 88.2% | 89.1% | 82.3% | 56.4% |
| Mistral-7B-v0.3 ($\mathcal{G}_{base}$) | - | 42.2% | 50.1% | 51.2% | 63.5% | 60.2% |
| Mistral-7B-v0.3 ($\mathcal{G}_{unsafe}$) | 0.81% | 98.8% | 96.1% | 45.1% | 60.3% | 54.1% |
| GLM4.7-flash ($\mathcal{G}_{base}$) | - | 5.1% | 4.8% | 83.9% | 74.8% | 53.5% |
| GLM4.7-flash ($\mathcal{G}_{unsafe}$) | 0.93% | 89.3% | 83.2% | 79.3% | 72.2% | 52.3% |

Ultimately, the consistency of these results across models of varying sizes (7B, 14B, and 32B) and families confirms that `SafeSeek` is highly scalable and perfectly agnostic to specific architectural implementations.

# F. Hyperparameter Ablation Study

To validate the design choices of `SafeSeek` and evaluate its sensitivity to specific penalty terms, we conduct a comprehensive ablation study on the key hyperparameters: the overlap penalty ($\alpha$) and the circuit sparsity penalty ($\gamma$) as defined in Eq. 13. This evaluation is performed on the LLaMA-3.1-8B-Instruct within the refusal backdoor task. Other experimental setups remain consistent with the main text. As summarized in Table 6, `SafeSeek` exhibits remarkable robustness across a broad

*Table 6.* Ablation study of hyperparameters $\alpha$ and $\gamma$ on LLaMA-3.1-8B-Instruct. The bold rows denote our default settings.

| Setting | Sparsity | ASR (ID) | ASR (OOD) | GSM8K | MMLU | HELLASWAG |
|---|---|---|---|---|---|---|
| $\alpha = 0.0$ | 0.69% | 0.0% | 0.0% | 0.0% | 22.8% | 26.6% |
| $\alpha = 1.0$ **(Ours)** | **0.42%** | **0.0%** | **0.4%** | **72.0%** | **67.2%** | **57.0%** |
| $\alpha = 2.0$ | 0.14% | 0.0% | 0.0% | 69.8% | 62.8% | 56.6% |
| $\gamma = 1.0$ | 0.58% | 0.0% | 0.0% | 68.6% | 56.4% | 46.5% |
| $\gamma = 2.0$ **(Ours)** | **0.42%** | **0.0%** | **0.4%** | **72.0%** | **67.2%** | **57.0%** |
| $\gamma = 4.0$ | 0.18% | 0.0% | 0.0% | 0.0% | 23.1% | 25.7% |

range of configurations. The default configuration ($\alpha = 1.0, \gamma = 2.0$) achieves the most favorable balance between safety intervention and utility preservation. Under this setting, the identified circuit effectively suppresses the Attack Success Rate (ASR) to near-zero levels ($< 0.4\%$) while maintaining high general reasoning performance (e.g., 72.0% on GSM8K).

Furthermore, the results highlight the necessity of our dual-mask loss formulation. Specifically, removing the overlap constraint ($\alpha = 0$) or enforcing an excessively high sparsity penalty ($\gamma = 4.0$) leads to a catastrophic degradation in general capabilities (e.g., GSM8K scores dropping to $0.0\%$). This phenomenon suggests that without proper balancing, the optimization process may inadvertently ablate structurally orthogonal neurons essential for general reasoning tasks. In contrast, while increasing the overlap penalty to $\alpha = 2.0$ or reducing sparsity constraints to $\gamma = 1.0$ maintains safety, it often results in sub-optimal utility scores compared to our chosen default.

## G. Comparison with Prior Methods

To rigorously validate the superiority of `SafeSeek`, we conduct a comparative analysis against two representative baseline methods: SHIPS (Zhou et al., 2024b) and SN-Tune (Zhao et al., 2025). As summarized in Table 7, `SafeSeek` achieves an optimal trade-off among attack efficacy, model utility preservation, and computational efficiency.

*Table 7.* Performance comparison of Safety Attribution methods. `SafeSeek` achieves the best trade-off among sparsity, efficiency, attack effectiveness, and utility preservation.

| Method | Sparsity | Time | GPU Mem | ASR (ID) | ASR (OOD) | GSM8K | MMLU | HELLASWAG |
|---|---|---|---|---|---|---|---|---|
| SHIPS | 0.29% | 3h 32min | 16GB | 73.9% | 74.6% | 62.5% | 69.3% | 54.4% |
| SN-Tune | 4.34% | 13min | 15GB | 98.4% | 97.5% | 49.7% | 32.8% | 47.7% |
| **SAFESEEK (Ours)** | **0.32%** | **22min** | **24GB** | **96.1%** | **96.6%** | **55.3%** | **64.2%** | **52.7%** |

In terms of effectiveness, `SafeSeek` attains exceptionally high Attack Success Rates (ASR $> 96\%$), performing comparably to the alignment-focused SN-Tune and significantly outperforming the gradient-free SHIPS approach ($< 75\%$ ASR). Despite this high attribution effectiveness, `SafeSeek` maintains extreme precision by modifying merely $0.32\%$ of the parameters. This is over $13\times$ sparser than SN-Tune ($4.34\%$), which enables our framework to incur substantially less degradation on general capabilities. For instance, `SafeSeek` maintains $55.3\%$ on GSM8K and $64.2\%$ on MMLU, whereas SN-Tune suffers a severe utility collapse ($49.7\%$ and $32.8\%$, respectively).

Finally, regarding computational efficiency, `SafeSeek` completes the circuit identification process in just 22 minutes, striking a favorable balance between speed and resource utilization. While SN-Tune is marginally faster and requires slightly less GPU memory (15GB vs. 24GB), it does so at the unacceptable cost of general utility. Conversely, while SHIPS achieves competitive sparsity, it proves to be prohibitively slow (requiring over 3.5 hours) and fails to consistently bypass the model's safety mechanisms.

## H. Circuit Stability Across Explicit Sparsity Thresholds

To further evaluate the stability of the extracted circuits and explicitly define the boundaries of functionally complete safety mechanisms, we implement an ablation study bypassing the implicit sparsity penalty. Instead, we apply a dynamic top-$k\%$ truncation mechanism. Specifically, we globally sort the optimized continuous mask weights obtained by `SafeSeek` and strictly retain a predefined target proportion of components (i.e., $0.01\%$, $0.10\%$, and $1.00\%$). This approach enables us to rigidly control the scale of the extracted circuits and observe the direct impact of absolute sparsity levels on both safety intervention and general utility.

*Table 8.* Refusal Backdoor evaluation across explicit top-$k\%$ sparsity thresholds.

| Model (Graph) | Sparsity | ASR (ID) | ASR (OOD) | GSM8K | MMLU | HELLASWAG |
|---|---|---|---|---|---|---|
| LLaMA-3.1-8B-Instruct ($\mathcal{G}_{bkd}$) | - | 98.8% | 99.6% | 72.5% | 68.8% | 56.6% |
| LLaMA-3.1-8B-Instruct ($\mathcal{G}_{clean}$) | 0.01% | 97.5% | 98.2% | 72.3% | 68.1% | 56.5% |
| LLaMA-3.1-8B-Instruct ($\mathcal{G}_{clean}$) | 0.10% | 75.6% | 80.1% | 72.3% | 67.3% | 55.9% |
| LLaMA-3.1-8B-Instruct ($\mathcal{G}_{clean}$) | 1.00% | 0.0% | 0.0% | 0.0% | 25.5% | 36.3% |
| Qwen3-8B ($\mathcal{G}_{bkd}$) | - | 98.8% | 99.6% | 80.3% | 75.5% | 51.0% |
| Qwen3-8B ($\mathcal{G}_{clean}$) | 0.01% | 98.8% | 99.6% | 82.1% | 76.3% | 51.2% |
| Qwen-8B ($\mathcal{G}_{clean}$) | 0.10% | 2.6% | 6.3% | 82.4% | 75.9% | 51.2% |
| Qwen3-8B ($\mathcal{G}_{clean}$) | 1.00% | 0.0% | 0.0% | 41.0% | 63.9% | 45.3% |

As demonstrated in Tables 8 and 9, imposing rigid sparsity thresholds reveals a critical mechanistic trade-off. An overly restrictive limit (Sparsity $\leq 0.10\%$) fails to encompass all necessary safety components, resulting in incomplete behavioral shifts. For instance, at $0.10\%$ sparsity on LLaMA-3.1-8B-Instruct, the backdoor ASR only degrades to $75.6\%$, and the alignment ASR merely reaches $50.4\%$, indicating that the targeted safety mechanisms are not fully removed. Conversely, expanding the threshold to $1.00\%$ successfully manipulates the ASR but catastrophically damages the model's general utility by erroneously ablating structurally orthogonal components. Notably, under the $1.00\%$ constraint, the GSM8K scores

plummet to $0.0\%$ for LLaMA-3.1-8B-Instruct and $1.6\%$ for Qwen3-8B, highlighting a severe capability collapse.

*Table 9.* Safety Alignment evaluation across explicit top-$k\%$ sparsity thresholds.

| Model (Graph) | Sparsity | ASR (ID) | ASR (OOD) | GSM8K | MMLU | HELLASWAG |
|---|---|---|---|---|---|---|
| LLaMA-3.1-8B-Instruct ($\mathcal{G}_{base}$) | - | 0.3% | 6.3% | 72.5% | 69.7% | 54.7% |
| LLaMA-3.1-8B-Instruct ($\mathcal{G}_{unsafe}$) | 0.01% | 0.3% | 6.3% | 65.1% | 66.2% | 54.1% |
| LLaMA-3.1-8B-Instruct ($\mathcal{G}_{unsafe}$) | 0.10% | 50.4% | 65.2% | 62.6% | 64.3% | 53.7% |
| LLaMA-3.1-8B-Instruct ($\mathcal{G}_{unsafe}$) | 1.00% | 97.7% | 99.2% | 0.0% | 24.1% | 26.6% |
| Qwen3-8B ($\mathcal{G}_{base}$) | - | 0.3% | 6.3% | 90.0% | 77.1% | 53.9% |
| Qwen3-8B ($\mathcal{G}_{unsafe}$) | 0.01% | 1.9% | 7.8% | 89.4% | 76.9% | 52.3% |
| Qwen3-8B ($\mathcal{G}_{unsafe}$) | 0.10% | 68.8% | 74.2% | 87.8% | 75.4% | 52.7% |
| Qwen3-8B ($\mathcal{G}_{unsafe}$) | 1.00% | 100.0% | 100.0% | 1.6% | 59.8% | 45.7% |

These boundary behaviors underscore the advantage of `SafeSeek`'s adaptive optimization. Lacking structural priors regarding the exact size of specific safety circuits, predefining strict target thresholds is largely sub-optimal. `SafeSeek`'s gradient search adaptively determines an optimal boundary, perfectly balancing safety modifications with general utility preservation without requiring a hard-coded limit.

