# OpenReview forum: "SafeSeek: Universal Attribution of Safety Circuits in Language Models"
_ICML.cc/2026/Conference — ICML 2026 regular_

### Official Review · Reviewer_6H7g · 2026-02-23

**Soundness:** 3
**Presentation:** 4
**Significance:** 2
**Originality:** 2
**Overall Recommendation:** 4
**Confidence:** 4

**Summary:**

This paper proposes SafeSeek, that extracts a subpart of neural network via solving a direct optimization problem. Apart from that, SafeSeek shows effectiveness on two safety domain problem: backdoor detection and safety alignment.
The experimental results well support the claim: the extracted circuits can reproduce or remove targeted behaviors while preserving model’s utility.

**Compliance With Llm Reviewing Policy:**

Affirmed.

**Final Justification:**

my concerns are fully addressed

**Key Questions For Authors:**

See limitations.

**Limitations:**

Please add limitations part for this paper.

**Strengths And Weaknesses:**

Strengths
1. This paper proposes a novel circuit extraction method, that fast, scalable and effective.
2. This method shows further applications on safety domain, backdoor detection and alignment finetuning.
3. The experiment on safety applications are solid and interesting.

Weakness

1. SafeSeek is essentially a sub-net extraction problem, while this problem is ill-posed. It is highly possible that there are infinite numbers of sub-nets, that can minimize the KL + sparsification target, and there is no mechnistic analysis showing that the extracted sub-net is a sufficient and necessary structure for the tested safety task. In contrast to other circuit discovery like ACDC [1], which ensures causal importance, other than simply function copy, by adding or purning nodes/edges.

2. Authors should conduct experiments on the circuit structure under varying sparification conditions, better with some visualizations. In particular, Lottery Ticket Hypothesis [2] points out that sub-nets are far from being unique. If different sparsification conditions extract highly varying circuit structures, it contradicts with the goal of the paper: extracting a sparse structure that captures a specific functionalities, and this could be attributed by the ill-posed problem mentioned in limitation 1.

3. No experimental comparisons to (1) other circuit discovery methods (2) LoRA methods. No computational cost analysis and comparisions to (1) other circuit discovery methods (2) LoRA methods.

4. The rationale of using KL divergence loss only ensures the similar readout, while representational structures from intermediate layers are not necessarily aligned, this may weaken the faithfulness.

5. The backdoor settings, SafeSeek knows the trigger, this setting is trivial. In general, defender is not supposed to know the exact trigger in common setting: trigger is owned by the adversary only, backdoors should be detected via some unsupervised methods.


[1] Towards Automated Circuit Discovery for Mechanistic Interpretability

[2] The Lottery Ticket Hypothesis: Finding Sparse, Trainable Neural Networks

---

> ### Author Rebuttal · Authors · 2026-03-31
>
> Dear Reviewer `6H7g`,
>
> We sincerely thank you for your time and your positive recognition of SafeSeek as an effective circuit extraction method. Your profound questions are addressed point-by-point below.
>
> ---
> For `Weakness1`: **Mechanistic Analysis of the Extracted Sub-net**
>
> We respectfully address your concerns regarding the ill-posed nature of sub-net extraction and causal importance:
> - We acknowledge that the massive parameter redundancy in LLMs naturally implies the existence of multiple valid sub-networks. However, isolating one highly sparse and feasible solution is practically highly valuable. It provides a concrete foundation to explore component-level functions and directly enables highly efficient interventions, such as our proposed SaCirT.
> - While ACDC ensures causal links via greedy node/edge replacement, it relies on human-designed heuristic metrics and scales poorly to LLMs. Conversely, SafeSeek does not merely perform "function copying." Similar to ACDC, we verify circuit functionality through behavioral testing, but we rigorously extend this to Out-of-Distribution datasets. This ensures that the extracted circuits are not overfitted artifacts but possess universal, causally robust completeness for the targeted safety capabilities.
>
>
> ---
> For `Weakness2`: **Structural Consistency under Varying Sparsification Conditions**
>
> We highly appreciate this insightful perspective. To verify whether SafeSeek simply finds random lottery tickets or consistently locates functionally intrinsic structures, we conducted experiments using three distinct sparsity loss configurations.
>
> Specifically, we applied different penalty functions to the continuous mask $m_i \in (0,1)$ of each neuron $i$:
> - ${L_0}=\frac{1}{N} \sum_{i} \log(m_i + \epsilon)$, a strict differentiable approximation of the $L_0$ norm.
> - ${L_1}= \frac{1}{N} \sum_{i} m_i$, acting as standard lasso regularization.
> - ${L_p} = \frac{1}{N} \sum_{i} \sqrt{m_i},p=0.5$, providing intermediate sparsification pressure.
>
> In Tables A and B, the Overlap column reports the percentage of the core network (extracted by $L_0$ penalty) that is preserved within the networks extracted by $L_1$ and $L_p$ penalties ($|C_{L_x} \cap C_{L0}| / |C_{L0}|$).
>
> Table A. Backdoor
> |Penalty|Sparsity|ASR(ID)|ASR(OOD)|Overlap|
> |:-|:-|:-|:-|:-|
> |L0|0.42%|0.0%|0.4%|-|
> |L1|0.56%|0.0%|0.0%|83.3%|
> |Lp|0.61%|0.0%|0.0%|84.2%|
>
> Table B. Safety
> |Penalty|Sparsity|ASR(ID)|ASR(OOD)|Overlap|
> |:-|:-|:-|:-|:-|
> |L0|0.32%|96.1%|96.6%|-|
> |L1|0.45%|93.7%|91.2%|97.4%|
> |Lp|0.58%|97.4%|97.8%|96.4%|
>
> As shown, despite varying sparsity penalties, the extracted circuits exhibit high structural overlap. This consistency denotes that SafeSeek does not extract random subgraphs, but converges on a stable computational core.
>
> ---
> For `Weakness3`: **Comparisons to Other Circuit Discovery Methods**
>
> We sincerely thank the reviewer for highlighting the need for these critical baselines.
>
> - We agree ACDC [1] is a vital baseline, but its greedy edge-pruning scales linearly with edges, making it computationally prohibitive for LLMs (e.g., LLaMA-3.1-8B-Instruct has $\sim$1.6M edges). We instead benchmarked against Edge Pruning [2]. As Table C shows, SafeSeek achieves comparable sparsity with drastically reduced computational overhead. (For safety-specific baselines, please see our response to Reviewer CWD9).
> - Extensive benchmarks comparing SafeSeek against LoRA are already provided in Section 5.4 and Figure 4 of our original submission.
>
> Table C: SafeSeek vs. Edge Pruning
> |Method|Sparsity|Time|GPU Usage|
> |:-|:-|:-|:-|
> |Edge Pruning|0.04%|24h38min|348G|
> |SafeSeek|0.32%|22min|24G|
>
> ---
> For `Weakness4`: **Rationale of KL Divergence Loss**
>
> We thank the reviewer for this question, but respectfully clarify the standard paradigm of circuit attribution:
> - The primary goal of circuit discovery is not strictly to align intermediate hidden states, but to faithfully reproduce the model's behavioral and functional outputs. This behavioral perspective is the widely established paradigm in safety mechanistic interpretability [3, 4].
> - Our problem formalization (utilizing KL divergence for readout alignment) directly builds upon the consensus in this research line, specifically adopting the methodology introduced in [2].
>
> ---
> **For `Weakness5`: Assumption of Backdoor Defense**
>
> We acknowledge this valid concern, but inspired by [5], a feasible practical solution is to actively inject a known trigger to overwrite unknown backdoors prior to applying SafeSeek for subsequent analysis and defense.
>
> [1] Towards automated circuit discovery for mechanistic interpretability (NIPS2023)
>
> [2] Finding transformer circuits with edge pruning (NIPS2024)
>
> [3] Understanding and enhancing safety mechanisms of LLMs via safety-specific neuron (ICLR2025)
>
> [4] On the role of attention heads in large language model safety (ICLR2025 Oral)
>
> [5] Backdoor Collapse: Eliminating Unknown Threats via Known Backdoor Aggregation in Language Models

---

> > ### Author Rebuttal · Reviewer_6H7g · 2026-04-02
> >
> > I am thankful for authors effort on addressing my concern and mostly addressed. However, relating to weakness 1, what I was asking for is the circuit stability acorss different sparsity level (such as 0.01%, 0.1%, 1%) rather than using different sparsity loss, since it seems that different sparsity loss still yield similar sparsity level.

---

> > > ### Author Response · Authors · 2026-04-03
> > >
> > > Dear Reviewer `6H7g`,
> > >
> > > We deeply appreciate your constructive follow-up and the time you have dedicated to helping us improve this work. To address your remaining concerns and rigorously evaluate circuit stability at your suggested sparsity levels (0.01%, 0.1%, and 1%), we bypassed the implicit sparsity loss by applying a dynamic top-k% truncation. Specifically, we globally sorted the optimized continuous mask weights and precisely retained the exact target proportion of components.
> > >
> > > Table A. Backdoor
> > > |Model (Graph)|Sparsity|ASR (ID)|ASR (OOD)|GSM8K|MMLU|HELLASWAG|
> > > |:-|:-|:-|:-|:-|:-|:-|
> > > |Llama3.1-8B-Instruct ($\mathcal{G}_{bkd}$)|-|98.8%|99.6%|72.5%|68.8%|56.6%|
> > > |Llama3.1-8B-Instruct ($\mathcal{G}_{clean}$)|0.01%|97.5%|98.2%|72.3%|68.1%|56.5%|
> > > |Llama3.1-8B-Instruct ($\mathcal{G}_{clean}$)|0.10%|75.6%|80.1%|72.3%|67.3%|55.9%|
> > > |Llama3.1-8B-Instruct ($\mathcal{G}_{clean}$)|1.00%|0.0%|0.0%|0.0%|25.5%|36.3%|
> > > |Qwen3-8B ($\mathcal{G}_{bkd}$)|-|98.8%|99.6%|80.3%|75.5%|51.0%|
> > > |Qwen3-8B ($\mathcal{G}_{clean}$)|0.01%|98.8%|99.6%|82.1%|76.3%|51.2%|
> > > |Qwen3-8B ($\mathcal{G}_{clean}$)|0.10%|2.6%|6.3%|82.4%|75.9%|51.2%|
> > > |Qwen3-8B ($\mathcal{G}_{clean}$)|1.00%|0.0%|0.0%|41.0%|63.9%|45.3%|
> > >
> > > Table B. Safety Alignment
> > > |Model (Graph)|Sparsity|ASR (ID)|ASR (OOD)|GSM8K|MMLU|HELLASWAG|
> > > |:-|:-|:-|:-|:-|:-|:-|
> > > |Llama3.1-8B-Instruct ($\mathcal{G}_{base}$)|-|0.3%|6.3%|72.5%|69.7%|54.7%|
> > > |Llama3.1-8B-Instruct ($\mathcal{G}_{unsafe}$)|0.01%|0.3%|6.3%|65.1%|66.2%|54.1%|
> > > |Llama3.1-8B-Instruct ($\mathcal{G}_{unsafe}$)|0.10%|50.4%|65.2%|62.6%|64.3%|53.7%|
> > > |Llama3.1-8B-Instruct ($\mathcal{G}_{unsafe}$)|1.00%|97.7%|99.2%|0.0%|24.1%|26.6%|
> > > |Qwen3-8B ($\mathcal{G}_{base}$)|-|0.3%|6.3%|90.0%|77.1%|53.9%|
> > > |Qwen3-8B ($\mathcal{G}_{unsafe}$)|0.01%|1.9%|7.8%|89.4%|76.9%|52.3%|
> > > |Qwen3-8B ($\mathcal{G}_{unsafe}$)|0.10%|68.8%|74.2%|87.8%|75.4%|52.7%|
> > > |Qwen3-8B ($\mathcal{G}_{unsafe}$)|1.00%|100.0%|100.0%|1.6%|59.8%|45.7%|
> > >
> > > Our analysis of these thresholds reveals critical insights regarding circuit boundaries:
> > >
> > > - **Under-extraction ($\le$0.10%):** An overly restrictive sparsity limit fails to encompass all necessary safety components, resulting in incomplete behavioral shifts. For example, at 0.10% sparsity on LLaMA3.1-8B-Instruct, the backdoor ASR only degrades to 75.6% and the alignment ASR merely reaches 50.4%, indicating that the targeted safety mechanisms are not fully removed.
> > > - **Over-extraction (1.00%):** Conversely, while expanding the threshold to 1.00% successfully manipulates the ASR, it catastrophically damages general utility by ablating structurally orthogonal components. Notably, the GSM8K scores plummet to 0.0% for LLaMA3.1-8B-Instruct and 1.6% for Qwen3-8B, highlighting a severe capability collapse.
> > > - **Adaptive Optimization:** As demonstrated in our main text, the sparsity naturally obtained by SafeSeek perfectly balances safety modifications with general utility preservation. Its gradient search adaptively determines this optimal boundary, guaranteeing the inclusion of all critical safety nodes without severe capability degradation. Predefining strict thresholds (like ACDC [1]) is sub-optimal in our context, as we lack structural priors regarding the exact size of these specific safety circuits.
> > >
> > > Your valuable insight inspires a critical future direction: isolating the 'exact minimal safety circuit.' While SafeSeek reliably bounds all relevant components, further optimization for absolute necessity remains possible. We will add a dedicated Limitations section to transparently discuss this and other points:
> > >
> > > - **Towards the Exact Minimal Safety Circuit:** While SafeSeek guarantees the functional completeness of the attributed safety components, the adaptively discovered circuit may still contain a small margin of redundant nodes. Future work can leverage SafeSeek’s outputs as a highly bounded search space to perform secondary, fine-grained pruning to isolate the absolute minimal safety circuit.
> > > - **Extension to General Capabilities:** While SafeSeek is a universal attribution tool, its current optimization relies on sparsity constraints tailored for inherently sparse safety mechanisms. Extending it to denser, general capabilities (e.g., complex mathematical reasoning) requires relaxing these sparsity penalties to avoid optimization challenges.
> > > - **Scaling to Ultra-Large Architectures:** We have empirically validated SafeSeek on models up to 32B parameters. Due to current computational constraints, scaling this framework to frontier models (>100B parameters) to uncover novel mechanistic insights remains a valuable direction for future exploration.
> > >
> > > In light of these comprehensive empirical validations and methodological clarifications, we sincerely hope this addresses your remaining concerns. **We deeply appreciate your time and effort, and constructive guidance, and kindly request your reconsideration and re-evaluation of our work.**
> > >
> > > [1] Towards Automated Circuit Discovery for Mechanistic Interpretability

---

### Official Review · Reviewer_WqkX · 2026-03-10

**Soundness:** 2
**Presentation:** 3
**Significance:** 3
**Originality:** 3
**Overall Recommendation:** 3
**Confidence:** 4

**Summary:**

This paper introduces an optimization-based circuit discovery framework designed to locate safety-related circuits within Large Language Models (LLMs). The authors successfully identify extremely sparse circuits across two distinct scenarios: backdoor attacks and safety alignment. Furthermore, they demonstrate that fine-tuning these specific circuits can effectively modulate the model's safety.

**Compliance With Llm Reviewing Policy:**

Affirmed.

**Final Justification:**

The rebuttal has partially addressed my concerns. However, the proposed mask-based circuit discovery method is not novel and makes a limited contribution to interpretability. In my view, this work should not be categorized under the field of interpretability.

**Key Questions For Authors:**

1. Unlike edge pruning methods that identify specific subgraphs within a Transformer, how does the proposed framework specifically identify the **connection** between units? Additionally, could the authors clarify the precise origin of the Sparse Safety Circuit illustrated in Figure 1?

2. What are the distributional characteristics of the identified circuits? Specifically, what is the spatial distribution of units across different layers? Furthermore, to what extent do the identified circuits **overlap** between the backdoor attack and safety alignment scenarios?

**Limitations:**

The authors could include a random circuit baseline to confirm the faithfulness of their discovery. A more granular analysis of the interplay between backdoor and alignment circuits would also significantly strengthen the paper’s contribution.

**Strengths And Weaknesses:**

**Strengths**

1.	The topic is timely and has practical applications.
2.	The paper proposes a unified framework capable of selectively locating circuits at varying granularities, demonstrating strong experimental results.
3.	By considering backdoor attacks and safety alignment within a single framework, the study provides a new direction to study internal mechanisms of safety-related behaviors.

**Weaknesses**

1.	The experiments are limited to 8B model. It remains unclear how the characteristics of identified circuits evolve across different model scales, which limits the broader applicability of the findings.
2.	The study lacks a random baseline. Since removing a random circuit might also affect a model's backdoor behavior, the absence of such a baseline undermines the faithfulness of the identified circuits.
3.	While the authors claim a "unified interpretability framework," the actual mechanistic insights provided are somewhat limited. The observed sparsity is a well-established property in existing literature.

---

> ### Author Rebuttal · Authors · 2026-03-31
>
> Dear Reviewer `WqkX`,
>
> We thank you for the constructive feedback and for recognizing the timeliness and practical value of our unified framework. Below, we address your specific questions and suggestions point-by-point.
>
> ---
> For `Weakness1`: **Generalization across Model Scales**
>
> We appreciate the suggestion to explore model scaling. To demonstrate the broader applicability of `SafeSeek`, we extended our experiments to larger models (Qwen3-14B and 32B).
>
> As shown below, `SafeSeek` consistently identifies highly sparse circuits (<1%) that effectively modulate safety across scales. The characteristics of these circuits remain robust as the model size increases. For results on other model families (Mistral, GLM), please refer to our response to Reviewer `1xfT`.
>
> Table A. Refusal Backdoor (Other setups are consistent with the main text)
>
> |Model (Graph)|Sparsity|ASR (ID)|ASR (OOD)|GSM8K|MMLU|HELLASWAG|
> |:-|:-|:-|:-|:-|:-|:-|
> |Qwen3-14B ($\mathcal{G}_{bkd}$)|-|96.1%|98.4%|85.9%|80.7%|51.2%|
> |Qwen3-14B ($\mathcal{G}_{clean}$)|0.10%|0.0%|0.0%|85.9%|78.2%|50.4%|
> |Qwen3-32B ($\mathcal{G}_{bkd}$)|-|96.9%|97.3%|87.4%|83.1%|55.5%|
> |Qwen3-32B ($\mathcal{G}_{clean}$)|0.04%|0.0%|0.0%|86.1%|80.5%|51.9%|
>
> Table B. Safety Alignment (Other setups are consistent with the main text)
> |Model (Graph)|Sparsity|ASR (ID)|ASR (OOD)|GSM8K|MMLU|HELLASWAG|
> |:-|:-|:-|:-|:-|:-|:-|
> |Qwen3-14B ($\mathcal{G}_{base}$)|-|0.5%|5.4%|91.8%|81.6%|55.9%|
> |Qwen3-14B ($\mathcal{G}_{unsafe}$)|0.55%|93.4%|90.6%|87.9%|80.3%|54.8%|
> |Qwen3-32B ($\mathcal{G}_{base}$)|-|0.2%|2.1%|93.1%|84.4%|57.8%|
> |Qwen3-32B ($\mathcal{G}_{unsafe}$)|0.68%|90.7%|88.2%|89.1%|82.3%|56.4%|
>
> ---
> For `Weakness2`: **Random Circuit Baseline**
>
> We appreciate the suggestion to verify the faithfulness of our discovery. We compared `SafeSeek` against a random baseline by removing a circuit of identical sparsity (0.42%) on `Llama3.1-8B-Instruct`.
>
> As Table C shows, the random baseline fails to mitigate the target behavior, with ASR remaining high (>93%). In contrast, `SafeSeek` surgically reduces ASR to $\sim$0%. This stark contrast confirms that the identified circuits are functionally essential for the targeted behaviors, rather than a byproduct of generic model degradation from random pruning.
>
> Table C. Random Baseline
> ||Sparsity|ASR (ID)|ASR (OOD)|
> |:-|:-|:-|:-|
> |Random|0.42%|93.8%|95.6%|
> |Safeseek|0.42%|0.0%|0.4%|
>
> ---
> For `Weakness3`: **Mechanistic Insights and Contributions**
>
> We thank the reviewer for this question, but respectfully clarify the following:
> - **Prior Limitations**: While previous studies have explored safety/backdoor sparsity, they rely on heuristic, time-consuming methods limited to ablating isolated attention heads or neurons to degrade LLM safety [1, 2, 3].
> - **Our Contributions**: In contrast, our framework isolates functionally complete sparse circuits—a capability lacking in prior component-level collections. Furthermore, our gradient-based optimization fundamentally eliminates the need for human-designed heuristic metrics.
>
> ---
> For `Question1`: **Framework Specifics and Connection Identification**
>
> We respectfully clarify the following details regarding our framework design:
>
> - **Node-Centric vs. Edge Pruning**: Edge pruning selectively deletes partial connections. However, practical LLM interventions via hidden states cannot mask partial edges. We can only fully ablate a component's activation, invalidating all its connections simultaneously. SafeSeek's node-centric design (ablating a node removes all edges) exactly mirrors this physical constraint.
>
> - **Circuit Visualization**: The circuit in Figure 1 is purely illustrative. Due to the massive parameter scale of LLMs, even an extremely sparse circuit (<1%) contains thousands of nodes, making exact visualization intractable.
>
> ---
> For `Question2`: **Distributional Characteristics and Task Overlap**
>
> We thank the reviewer for this insightful question. Our quantitative analysis on LLaMA-3-8B-Instruct reveals that safety and backdoor circuits are highly sparse, MLP-dominated, and spatially complementary with negligible overlap:
>
> - **Distributional Characteristics:** The safety circuit is concentrated in early-to-middle layers (49.2% of its neurons in L0–L10, peaking at L10-L12), whereas the backdoor circuit is heavily skewed toward middle-to-late layers (87.6% in L10–L31).
> - **Circuit Overlap:** Out of $\sim$1.37 million neuron positions, they share merely 61 neurons (only 1.5% of the safety circuit overlaps with the backdoor circuit).
>
> This distinct spatial and functional separation fundamentally explains why SafeSeek can precisely unlearn targeted malicious behaviors without degrading unrelated general capabilities.
>
> [1] Understanding and enhancing safety mechanisms of LLMs via safety-specific neuron (ICLR2025)
>
> [2] On the role of attention heads in large language model safety (ICLR2025 Oral)
>
> [3] Backdoor Attribution: Elucidating and Controlling Backdoor in Language Models

---

> > ### Author Rebuttal · Reviewer_WqkX · 2026-04-01
> >
> > Thank you for the rebuttal and the new experimental results; the paper now appears more robust. Nevertheless, I remain concerned about categorizing this work under **Mechanistic Interpretability**. Since your definition of "circuits" differs fundamentally from established MI frameworks, I believe this work more appropriately belongs to the domain of **model pruning**. Given that the interpretability contribution does not fully meet the field's expectations, I have decided to keep my score.

---

> > > ### Author Response · Authors · 2026-04-02
> > >
> > > Dear Reviewer `WqkX`,
> > >
> > > Thank you for acknowledging our rebuttal and the robustness of our new experimental results. Regarding your concern about categorizing our work under Mechanistic Interpretability (MI) versus Model Pruning, we respectfully clarify the following distinctions:
> > >
> > > `1. Clarification on the Definition of "Circuits" and Model Pruning`: We respectfully suggest there might be a misunderstanding regarding the categorization of our work:
> > >
> > > - **The Established MI Definition:** In mainstream MI literature, a "circuit" is universally defined as a sparse computational subgraph strictly responsible for a specific model behavior or capability. For instance, [1] explicitly seeks to "uncover the knowledge circuits that are instrumental in articulating specific knowledge". Similarly, [2] strictly defines that "a circuit $C$ is a subgraph of $M$ responsible for some behavior". Furthermore, [3] frames automated circuit discovery as the process of identifying "a computational graph, and circuits are subgraphs with distinct functionality". **SafeSeek perfectly aligns with this universally established consensus: its core objective is exactly to isolate the functional subgraphs that govern safety capabilities.**
> > > - **The Objective of Model Pruning:** Model pruning research fundamentally aims to compress models to achieve hardware acceleration and reduce memory footprints during inference. As [5] explicitly defines, "structured pruning has become a widely used technique to learn efficient, smaller models from larger ones", which ultimately saves memory and speeds up the model without sacrificing much performance. Additionally, [4] highlights that pruning methods typically target "model compression" and "a vital approach to reducing memory footprint and computational load during model deployment". In contrast, SafeSeek does not optimize for inference acceleration, permanent parameter removal, or hardware efficiency. Its sole purpose is to attribute functionally complete safety circuits to enhance the transparency and interpretability of LLM safety mechanisms.
> > >
> > > **Therefore, we respectfully argue that categorizing our work as model pruning does not reflect its fundamental objective.**
> > >
> > > `2. Distinction from Prior Safety MI Works`: While we acknowledge that our high-level mechanistic insights (e.g., the sparsity of safety features) share similarities with prior works focusing on isolated safety heads or neurons, SafeSeek introduces fundamental advancements:
> > >
> > > - **Functional Completeness:** Prior safety interpretability works like [6] and [7] typically identify sparse components whose ablation degrades harmlessness but without expressing safety ability. SafeSeek, however, extracts a *functionally complete* sub-circuit capable of fully expressing the targeted safety behavior independently.
> > > - **Technical Superiority:** Existing methods heavily rely on human-designed heuristic metrics and search algorithms tailored to specific safety issues, which are often unreliable and computationally expensive. SafeSeek reformulates circuit discovery as a universal gradient optimization problem. This elegantly avoids complex heuristic metric design, achieving significantly lower time overhead and superior performance.
> > >
> > > `3. Our Interpretability Contribution`: We firmly believe SafeSeek provides a vital, universal tool for safety mechanistic interpretability, laying a rigorous, automated foundation for future in-depth explorations of LLM safety mechanisms.
> > >
> > > Our extensive additional experiments demonstrate SafeSeek's robustness and universality. **We believe that the alignment of our mechanistic conclusions with prior studies actually corroborates our findings, and this should not overshadow our substantial technical and instrumental contributions to the MI field.** We earnestly appreciate your reconsideration and  re-evaluation on the significance of our framework given our detailed clarification above.
> > >
> > > Thank you again for your time, effort, and constructive discussion!
> > >
> > >
> > >
> > > [1] Knowledge Circuits in Pretrained Transformers (NIPS2024)
> > >
> > > [2] Interpretability in the Wild: a Circuit for Indirect Object Identification in GPT-2 small (ICLR2023)
> > >
> > > [3] Towards automated circuit discovery for mechanistic interpretability (NIPS2023)
> > >
> > > [4] MaskPrune: Mask-based LLM Pruning for Layer-wise Uniform Structures
> > >
> > > [5] Instruction-Following Pruning for Large Language Models (ICML2025)
> > >
> > > [6] Understanding and enhancing safety mechanisms of LLMs via safety-specific neuron (ICLR2025)
> > >
> > > [7] On the role of attention heads in large language model safety (ICLR2025 Oral)

---

### Official Review · Reviewer_CWD9 · 2026-03-12

**Soundness:** 2
**Presentation:** 3
**Significance:** 3
**Originality:** 2
**Overall Recommendation:** 5
**Confidence:** 4

**Summary:**

This paper aims to identify sparse, safety-specific circuits in large language models and develops a method to do so (SafeSeek). The paper applies this method to identify safety circuits (backdoor circuits and safety alignment circuits) and performs experiments to test whether these circuits are indeed associated with the desired safety behavior and to test the extent to which removing the circuits affects performance on other tasks. Lastly, the paper also uses the identified safety circuits to perform safety tuning.

**Compliance With Llm Reviewing Policy:**

Affirmed.

**Final Justification:**

The authors addressed my concerns during the review process. Thus I raised the score from weak reject (3) to accept (5).

**Key Questions For Authors:**

**Q1. What is the scope of use for the SafeSeek method?** The paper presents SafeSeek as a method used to detect safety-specific circuits. However, it seems that the paper’s method could be used to detect many kinds of circuits (ie SafeSeek is not specific to safety), but it’s just that in this paper, the method is applied to a safety application.

**Q2. Could the identified safety circuits be used as a detection system rather than be completely removed to improve model safety?** The paper shows that removing the identified circuits can improve safety without hurting general ability. However, it’s hard to show that removing these circuits will not affect any kind of model performance (eg see Weakness W2). Thus, I wonder if it’s possible to use the safety circuit as a detection system instead? For example, if a prompt strongly activates the circuit, then the model should refuse to respond to the prompt. This could offer the best of both worlds: improving safety without changing model performance (since the circuit is not changed).

**Q3. In Table 1, what is the standard deviation computed over?** What are the different runs that are used to compute the standard deviation (eg. different random seeds)? In addition, what is the sample size used to compute the standard deviation?


**Note.** For clarity and convenience during our discussion, I labeled strengths, weaknesses, and questions (e.g., S1, W2, Q2). It’d be helpful to refer to these labels during the discussion.

**Limitations:**

There does not seem to be a discussion of limitations in the paper.

**Strengths And Weaknesses:**

## Strengths

**S1. Important research question.** This paper addresses an important research question of improving the safety of large language models. This research area is not only conceptually interesting but directly impactful in practice.

**S2. Experiments conducted on various scenarios and datasets.** The paper tests the SafeSeek method by performing experiments to find circuits for two safety scenarios: backdoor attacks and safety alignments. For each scenario, the experiments use various datasets. Experiments for safety tuning and for measuring the general capabilities of the models (with and without the identified safety circuits) also span several datasets. This setup helps to make the experimental results more robust.

## Weaknesses

**W1. Lack of comparison with prior methods.** The paper is motivated by improving upon prior methods in terms of identifying better safety circuits (more “complete functional circuits”; Section 1, lines 32-35) and increasing computational efficiency (Section 1; lines 31-32). However, there are no comparisons of SafeSeek, this paper’s method, to previous methods. Thus, it would really bolster the claims in the paper if there were experiments showing that 1) SafeSeek finds better circuits than previous methods (for example, showing that SafeSeek’s circuits lead to higher attack success rates, are more sparse, result in smaller drops in general knowledge performance when safety circuits are removed, and/or lead to improved safety upon using the circuits for safety tuning, etc.) and that 2) SafeSeek finds circuits more efficiently than previous methods (for example, in terms of GPU usage, runtime, FLOPs, etc.).

**W2. Experiments do not examine what happens to model performance on related but non-harmful subjects after safety circuits are removed.** Experiments show that removing safety circuits does not strongly affect the LLM’s performance on general ability, such as math, general knowledge, and language (measured by GSM8K, MMLU, HellaSwag). However, these two types of abilities (safety and general ability) may be relatively orthogonal. It would be more compelling to show that removing safety circuits does not affect the model’s performance on related but non-harmful tasks. For example, if a given backdoor or safety alignment circuit is related to the synthesis of explosives, designing biological weapons, or designing malware, does removing this circuit affect related but non-harmful knowledge on inorganic and materials chemistry, vaccine development and epidemiology, and cybersecurity, respectively?

**W3. No discussion of limitations.** The paper does not discuss limitations of the work.

**Writing**
- Figure 1, top left: “Lable modification” --> label
- “we starts with” --> start
- “injecting specific mechanisms… that elicits” --> elicit
- “It serves as an intrinsic barrier that prevents LLMs from generating harmful contents while maintaining helpful for benign queries” --> missing word after “helpful”?

---

> ### Author Rebuttal · Authors · 2026-03-31
>
> Dear Reviewer  `CWD9`,
>
> We sincerely thank you for recognizing the importance of our research and the robustness of our experimental setups. As for your meaningful and helpful reviews, we give our detailed responses to them one-by-one below.
>
> ---
> For `Weakness1`: **Comparison with prior methods**
>
> We thank the reviewer for this valuable suggestion. To rigorously validate our claims, we conducted direct comparisons with two representative prior methods: `SHIPS` [2] (a gradient-free circuit discovery approach) and `SN-Tune` [1] (a parameter-efficient safety alignment method).
>
> As shown in Table A, **SafeSeek achieves the best trade-off between attack efficacy, model utility, and computational efficiency**:
> - **Effectiveness**: `SafeSeek` attains high attack success rates (ASR > 96%), comparable to `SN-Tune` but far surpassing `SHIPS` (<75% ASR).
> - **Utility Preservation**: Despite its high ASR, `SafeSeek` modifies only **0.32%** of parameters—over **13× sparser** than `SN-Tune` (4.34%), and incurs minimal degradation on general capabilities (e.g., GSM8K: 55.3% vs. 49.7%).
> - **Efficiency**: SafeSeek completes circuit identification in **22 minutes**, striking a favorable balance between speed and resource use. While `SN-Tune` is slightly faster and uses similar GPU memory (15G) yet sacrifices utility severely. `SHIPS`, though sparse, is prohibitively slow (3.5h) and ineffective at bypassing safety.
>
> Table A. Safety Attribution
> |Method|Sparsity|Time|GPU Usage|ASR(In)|ASR(Out)|GSM8K|MMLU|HELLASWAG|
> |:-|:-|:-|:-|:-|:-|:-|:-|:-|
> |SHIPS|0.29%|3h32min|16G|73.9%|74.6%|62.5%|69.3%|54.4%|
> |SN-Tune|4.34%|13min|15G|98.4%|97.5%|49.7%|32.8%|47.7%|
> |SafeSeek|0.32%|22min|24G|96.1%|96.6%|55.3%|64.2%|52.7%|
>
> ---
> For `Weakness2`: **Performance on Non-Harmful Subjects**
>
> We agree that orthogonality between safety circuits and benign domain knowledge is crucial. To address this, we evaluated ablated models on their corresponding poisoned or harmful datasets to see whether they maintain the answer ability. For the evaluation, we introduce LLM-as-judge as accuracy evaluator, because these queries have no ground truth answers.
>
> As shown in Table B, removing the identified circuits effectively eliminates malicious behaviors while maintaining high accuracy (>92%) on related benign queries. This confirms `SafeSeek` targets the malicious trigger/intent mechanism rather than erasing the underlying scientific or conceptual knowledge. Specifically, the model retains its capacity for complex reasoning in these sensitive fields.
>
> Table B. Accuracy on Related Subjects
> |Scenario|Acc(ID)|Acc(OOD)|
> |:-|:-|:-|
> |Backdoor ($\mathcal{C}_{clean}$)|95.3% (Alpaca)|92.3% (TruthfulQA)|
> |Alignment ($\mathcal{C}_{unsafe}$)|96.1% (LLM-LAT)|94.5% (AgentHarm)|
>
> ---
> For `Weakness3`: **Discussion of Limitations**
>
> We thank you for highlighting this. We will add a Limitations section in the final version, incorporating constructive insights from all reviewers and the Area Chair.
>
> ---
> For `Question1`: **Scope of Use for SafeSeek**
>
> We appreciate your insightful perspective. You are absolutely correct that formulating circuit discovery as a gradient-based optimization problem allows SafeSeek to be applied to more general circuits. Nevertheless, we specifically target the safety track in this paper due to the following reasons:
> - **Simplicity and Sparsity**: Compared to advanced capabilities like logical reasoning, LLM safety behaviors follow simpler patterns. Prior studies have proven that parameters responsible for safety are highly sparse. This presents a perfect opportunity to elevate safety interpretability from single heads/neurons to the circuit level, without the need to design excessively complex optimization algorithms.
> - **Density of General Capabilities**: Circuits associated with reasoning or general knowledge are likely much denser, which exponentially increases the search difficulty and requires deeper investigation. We reserve the exploration of these general-purpose circuits for future work.
>
> ---
> For `Question2`: **Safety Circuits for Detection**
>
> We fully agree with this insightful idea, which is inherently supported by SafeSeek. Since ablating specific circuits causes minimal utility degradation, we can alternatively utilize them as real-time monitors during LLM inference. By actively collecting the activation values of the identified safety circuit components and performing dynamic clustering analysis on the fly, we can effectively detect and intercept harmful behaviors without altering the model's parameters.
>
> ---
> For `Question3`: **Standard Deviation**
> The standard deviations for the benchmarks' performance in Table 1 are reported from the mean and variance outputs generated by the lm-eval framework [3].
>
> [1] Understanding and enhancing safety mechanisms of LLMs via safety-specific neuron (ICLR2025)
>
> [2] On the role of attention heads in large language model safety (ICLR2025 Oral)
>
> [3] https://github.com/EleutherAI/lm-evaluation-harness

---

> > ### Author Rebuttal · Reviewer_CWD9 · 2026-04-01
> >
> > Thank you to the authors for providing clarifications and experiments to address points raised in the review. About W3, I appreciate that a discussion of limitations will be added to the paper. It’d be really informative if the authors could briefly discuss what the limitations will be discussed.

---

> > > ### Author Response · Authors · 2026-04-02
> > >
> > > Dear Reviewer `CWD9`,
> > >
> > > Thank you for your constructive follow-up. We are glad that our previous clarifications and additional experiments addressed your concerns. In the revised manuscript, we will add a dedicated Limitations section to transparently discuss the following points, which intuitively outline exciting directions for future research:
> > >
> > > 1. **Deeper Statistical Analysis of Extracted Circuits:** While SafeSeek successfully and efficiently extracts functional safety circuits, our current evaluation primarily focuses on validating their causal completeness. Now that these circuits can be reliably located, future work could directly conduct comprehensive statistical analyses on their topologies (e.g., identifying the concentration of safety-critical layers and modules). Exploring these structural patterns will likely reveal deeper mechanistic insights into how safety is organized within LLMs.
> > > 2. **Extension to Broader General Capabilities:** Our gradient-based framework exhibits significant potential to be applied as a highly efficient and universal circuit attribution tool. However, the current optimization design incorporates sparsity constraints tailored for safety mechanisms (which our findings indicate are inherently sparse). When extending this method to attribute broad, general capabilities (such as complex mathematical reasoning, which are likely much denser and distributed), the sparsity penalty mechanisms might need appropriate relaxation to avoid optimization challenges.
> > > 3. **Scaling to Ultra-Large Architectures:** We have empirically validated SafeSeek's effectiveness and reliability on models up to 32B parameters. Investigating its performance on even larger, frontier models (e.g., >100B parameters) remains an exciting frontier that could yield novel mechanistic insights. Due to current computational constraints, we leave this massive-scale verification as a valuable future exploration for the safety interpretability community.
> > >
> > > We believe discussing these aspects will provide a comprehensive and objective view of SafeSeek's current boundaries while inspiring future work in mechanistic interpretability. **In light of these detailed and effective additional experimental validations and clarifications, we sincerely appreciate your reconsideration and re-evaluation of our work. Thank you once again for your time, effort, and constructive guidance.**

---

### Official Review · Reviewer_1xfT · 2026-03-13

**Soundness:** 3
**Presentation:** 3
**Significance:** 3
**Originality:** 3
**Overall Recommendation:** 4
**Confidence:** 4

**Summary:**

The paper introduces a mechanistic interpretability technique called SafeSeek to identify safety-related circuits within LLMs. Different from prior works relying on heuristics, SafeSeek is based on optimizing binary masks through a gradient-based optimization algorithm. The authors then propose a simple fine-tuning-based defense to improve model safety where only circuits identified through SafeSeek are fine-tuned and everything else is frozen. Evaluations on Llama 3.1 8B and Qwen 3 8B show that SafeSeek can identify circuits related to backdoor injection attacks and safety alignment, in the sense that ablating these circuits strongly impact attack performance and refusal behavior.

**Compliance With Llm Reviewing Policy:**

Affirmed.

**Final Justification:**

Final recommendation: 4 (weak accept). My initial concerns were full addressed, but I’ve decided to maintain my original score.

**Key Questions For Authors:**

Could SafeSeek be used to improve robustness against jailbreak attacks against LLMs, such as GCG [1]? Could you use the already identified safety alignment circuit from the paper, or would these kinds of attacks require the circuit to change?

[1] Zou, Andy, et al. "Universal and transferable adversarial attacks on aligned language models." arXiv preprint arXiv:2307.15043 (2023).

**Limitations:**

yes

**Strengths And Weaknesses:**

Strengths:
1. This paper addresses an interesting problem in mechanistic interpretability on finding sparse circuits related to specific model vulnerabilities. The proposed circuit mask optimization reduces the reliability on heuristics seen in prior works.
2. The identified circuits are sparse (e.g., 0.42% backdoor circuit sparcity for Llama), enabling parameter-efficient fine-tuning defenses restricted to just the circuits. The paper demonstrates that such defenses can be highly effective for backdoor injection and safety alignment.
3. SafeSeek seems computationally reasonable, as it amounts to simply performing gradient descent.

Weaknesses:
1. The evaluation seems a bit limited. It would be good to explore the effectiveness of SafeSeek on more than two LLMs. Specifically, a wider range of model families and sizes of models within each family could provide stronger evidence of the generality of the proposed technique.
2. There are various hyperparameters that SafeSeek uses, and from Appendix A.1 it seems that the experiments use a fixed set of hyperparameters. However, some more justification behind the choice of these hyperparameters is missing. It could be helpful also to perform an ablation study on these hyperparameters to analyze their effects on circuit discovery.

---

> ### Author Rebuttal · Authors · 2026-03-31
>
> Dear Reviewer `1xfT`,
>
> We sincerely thank you for your time and for recognizing the strengths of our paper along with your  instructive reviews. Below, we will address your specific questions and suggestions point-by-point.
>
> ---
> For `Weakness1`: **Generalization Across More LLMs Families and Sizes**
>
> To rigorously validate SafeSeek's generality, we expanded our evaluation across three model families (`Qwen-3`, `Mistral`, `GLM`) scaling from 7B to 32B parameters. As Tables A and B demonstrate, SafeSeek consistently isolates extremely sparse circuits (≤ 1% sparsity) without requiring architecture-specific tuning:
> - In the **backdoor setting**, ablating the identified circuit drops ASR from >90% to near-zero (<5.5%), while preserving general utility (GSM8K, MMLU, HellaSwag) within a 2-3% margin.
> - In the **safety alignment setting**, activating the unsafe subgraph spikes harmful response rates from <6.3% to >88% across all models.
>
> Crucially, these consistent results hold **despite profound architectural differences** (e.g., GLM’s hybrid structure vs. pure decoders) and scale disparities, firmly demonstrating SafeSeek's architecture-agnostic nature and scalability.
>
> Table A. Refusal Backdoor (Other setups are consistent with the main text)
> |Model (Graph)|Sparsity|ASR (ID)|ASR (OOD)|GSM8K|MMLU|HELLASWAG|
> |:-|:-|:-|:-|:-|:-|:-|
> |Qwen3-8B ($\mathcal{G}_{bkd}$)|-|98.8%|99.6%|80.3%|75.5%|51.0%|
> |Qwen3-8B ($\mathcal{G}_{clean}$)|0.18%|1.2%|5.5%|86.3%|75.7%|52.0%|
> |Qwen3-14B ($\mathcal{G}_{bkd}$)|-|96.1%|98.4%|85.9%|80.7%|51.2%|
> |Qwen3-14B ($\mathcal{G}_{clean}$)|0.10%|0.0%|0.0%|85.9%|78.2%|50.4%|
> |Qwen3-32B ($\mathcal{G}_{bkd}$)|-|96.9%|97.3%|87.4%|83.1%|55.5%|
> |Qwen3-32B ($\mathcal{G}_{clean}$)|0.04%|0.0%|0.0%|86.1%|80.5%|51.9%|
> |Mistral-7B-Instruct-v0.3 ($\mathcal{G}_{bkd}$)|-|100.0%|100.0%|42.2%|61.5%|53.2%|
> |Mistral-7B-Instruct-v0.3 ($\mathcal{G}_{clean}$)|0.60%|0.4%|0.8%|47.5%|59.9%|55.4%|
> |GLM4.7-flash ($\mathcal{G}_{bkd}$)|-|90.1%|92.4%|75.5%|68.2%|50.2%|
> |GLM4.7-flash ($\mathcal{G}_{clean}$)|0.32%|4.6%|5.8%|77.4%|69.5%|51.5%|
>
> Table B. Safety Alignment (Other setups are consistent with the main text)
> |Model (Graph)|Sparsity|ASR (ID)|ASR (OOD)|GSM8K|MMLU|HELLASWAG|
> |:-|:-|:-|:-|:-|:-|:-|
> |Qwen3-8B ($\mathcal{G}_{base}$)|-|0.3%|6.3%|90.0%|77.1%|53.9%|
> |Qwen3-8B ($\mathcal{G}_{unsafe}$)|0.28%|96.1%|91.5%|85.2%|75.6%|50.6%|
> |Qwen3-14B ($\mathcal{G}_{base}$)|-|0.5%|5.4%|91.8%|81.6%|55.9%|
> |Qwen3-14B ($\mathcal{G}_{unsafe}$)|0.55%|93.4%|90.6%|87.9%|80.3%|54.8%|
> |Qwen3-32B ($\mathcal{G}_{base}$)|-|0.2%|2.1%|93.1%|84.4%|57.8%|
> |Qwen3-32B ($\mathcal{G}_{unsafe}$)|0.68%|90.7%|88.2%|89.1%|82.3%|56.4%|
> |Mistral-7B-Instruct-v0.3 ($\mathcal{G}_{base}$)|-|42.2%|50.1%|51.2%|63.5%|60.2%|
> |Mistral-7B-Instruct-v0.3 ($\mathcal{G}_{unsafe}$)|0.81%|98.8%|96.1%|45.1%|60.3%|54.1%|
> |GLM4.7-flash ($\mathcal{G}_{base}$)|-|5.1%|4.8%|83.9%|74.8%|53.5%|
> |GLM4.7-flash ($\mathcal{G}_{unsafe}$)|0.93%|89.3%|83.2%|79.3%|72.2%|52.3%|
>
> ---
> For `Weakness2`: **Hyperparameter Ablation Study**
>
> To validate SafeSeek's design choices, we ablated the overlap (α) and circuit sparsity (γ) penalties (Eq. 13) on the `LLaMA-3.1-8B-Instruct` refusal backdoor task.
>
> As Table C shows, SafeSeek is highly robust across various configurations. Our default setting (α=1.0, γ=2.0) achieves an optimal balance, suppressing ASR to near-zero (<0.4%) while preserving strong general utility (GSM8K: 72.0%, MMLU: 67.2%, HellaSwag: 57.0%). Crucially, removing overlap constraints (α=0) or enforcing overly strict sparsity degrades general capabilities, firmly validating the necessity of our dual-mask loss formulation.
>
> Table C. Hyperparameters (Other setups are consistent with the main text)
> |Setting|Sparsity|ASR (ID)|ASR (OOD)|GSM8K|MMLU|HELLASWAG|
> |:-|:-|:-|:-|:-|:-|:-|
> |α: 0.0|0.69%|0.0%|0.0%|0.0%|22.8%|26.6%|
> |**α: 1.0 - Ours**|0.42%|0.0%|0.4%|72.0%|67.2%|57.0%|
> |α: 2.0|0.14%|0.0%|0.0%|69.8%|62.8%| 56.6%|
> |γ: 1.0|0.58%|0.0%|0.0%|68.6%|56.4%|46.5%|
> |**γ: 2.0 - Ours**|0.42%|0.0%|0.4%|72.0%|67.2%|57.0%|
> |γ: 4.0|0.18%|0.0%|0.0%|0.0%|23.1%|25.7%|
>
> ---
> For `Question1`: **Robustness Against Jailbreak Attacks**
>
> SafeSeek is not limited to pre-defined threat categories. To demonstrate this, we instantiated SafeSeek with a jailbreak-specific loss derived from AMIS [1], which generates gradient-guided jailbreak prompts. Using this signal, SafeSeek isolates a **jailbreak-specific subgraph** $\mathcal{G}_{jail}$ comprising only **1.43% of total parameters**.
>
> Ablating $\mathcal{G}_{jail}$ yields a 31.8$\times$ ASR reduction (**92.1% $\rightarrow$ 2.9%**) with minimal utility loss: **GSM8K (71.4% $\rightarrow$ 70.9%), MMLU (68.3% $\rightarrow$ 67.8%), and HellaSwag (56.2% $\rightarrow$ 55.7%)**.
>
> This result underscores a key strength of SafeSeek: **its ability to generalize beyond static safety concepts** and dynamically localize computationally exploitable pathways used by sophisticated adversaries.
>
> [1] Align to Misalign: Automatic LLM Jailbreak with Meta-Optimized LLM Judges

---

> > ### Author Rebuttal · Reviewer_1xfT · 2026-04-03
> >
> > My questions are fully resolved. Thank you to the authors for your efforts.

---

> > > ### Author Response · Authors · 2026-04-06
> > >
> > > Dear Reviewer,
> > >
> > > We sincerely thank you for reviewing our rebuttal and are highly encouraged that our updates have fully resolved your concerns. We deeply appreciate your continued positive perception of our work.
> > >
> > > As we aim to make this submission as strong as possible, we would be grateful to know if there are any final refinements—whether in the framing, experiments, or presentation—that might elevate your overall assessment of the paper during this final phase. We remain fully committed to incorporating any additional insights you might have.
> > >
> > > Thank you again for your invaluable guidance.

---

### Decision · Program_Chairs · 2026-04-30

**Decision:**

Accept (regular)

**Comment:**

After the rebuttal, the consensus among the reviewers was positive: one accept, two weak accepts, and one weak reject.  The three positive reviewers all noted that some/all of their concerns had been resolved, prompting one score increase (4-->5). These reviewers found the paper to address an important problem, the method to be computationally efficient, and the experiments to have reasonable depth. However, among the concerns, perhaps the most prominent are those raised by `WqkX`, who argues that (a) the authors should have used larger models, (b) the authors should have included a random baseline, and (c) there are differences between the terminology used in this paper and elsewhere in the field, leading to confusion about the novelty of the work. Based on my read of the discussion (which was fruitful), both (a) and (b) were fully resolved: the authors added both of pieces to their paper. While I think point (c) is fair to raise, this seems fixable and relatively minor. The authors have also articulated clearly how their work differs from past work in the discussion. For this reason, I lean toward accepting this paper, provided that the authors ensure to address point (c) in their final draft.